# CD83 expression characterizes precursor exhausted T cell population

Zhiwen Wu[1], Toshiaki Yoshikawa[1,6], Satoshi Inoue[1], Yusuke Ito[1,6], Hitomi Kasuya[1], Takahiro Nakashima[1,2], Haosong Zhang[1,3], Saki Kotaka[4], Waki Hosoda[5], Shiro Suzuki[4] & Yuki Kagoya [1,3,6 ✉]

T cell exhaustion is a main obstacle against effective cancer immunotherapy. Exhausted T cells include a subpopulation that maintains proliferative capacity, referred to as precursor exhausted T cells ($T_{PEX}$). While functionally distinct and important for antitumor immunity, $T_{PEX}$ possess some overlapping phenotypic features with the other T-cell subsets within the heterogeneous tumor-infiltrating T-lymphocytes (TIL). Here we explore surface marker profiles unique to $T_{PEX}$ using the tumor models treated by chimeric antigen receptor (CAR)-engineered T cells. We find that CD83 is predominantly expressed in the $CCR7^+PD1^+$ intratumoral CAR-T cells compared with the $CCR7^-PD1^+$ (terminally differentiated) and CAR-negative (bystander) T cells. The $CD83^+CCR7^+$ CAR-T cells exhibit superior antigen-induced proliferation and IL-2 production compared with the $CD83^-$ T cells. Moreover, we confirm selective expression of CD83 in the $CCR7^+PD1^+$ T-cell population in primary TIL samples. Our findings identify CD83 as a marker to discriminate $T_{PEX}$ from terminally exhausted and bystander TIL.

[1] Division of Immune Response, Aichi Cancer Center Research Institute, Nagoya, Japan. [2] Department of Hematology and Oncology, Nagoya City University Graduate School of Medical Sciences, Nagoya, Japan. [3] Division of Cellular Oncology, Department of Cancer Diagnostics and Therapeutics, Nagoya University Graduate School of Medicine, Nagoya, Japan. [4] Department of Gynecologic Oncology, Aichi Cancer Center, Nagoya, Japan. [5] Department of Pathology and Molecular Diagnostics, Aichi Cancer Center, Nagoya, Japan. [6] Present address: Division of Tumor Immunology, Institute for Advanced Medical Research, Keio University School of Medicine, Tokyo, Japan. ✉email: ykagoya@keio.jp

Cytotoxic CD8$^+$ T cells that recognize tumor antigens migrate to the tumor microenvironment to induce a potent immune response. However, tumor-infiltrating T lymphocytes (TILs) are persistently exposed to the target antigen and rendered "exhausted." Exhausted T cells have attenuated proliferative capacity, cytokine production, and cytolytic activity. Although initially reported in the context of chronic viral infections, T cell exhaustion is also induced in the tumor microenvironment[1–3]. Recent studies analyzing the molecular signatures of T cells at the single-cell level have demonstrated that intratumoral exhausted T cells are not a homogeneous population but comprise cells at varying differentiation statuses[4,5]. T cells that are functionally exhausted but share gene expression profiles with early memory T cells, including the transcription factor TCF7 and surface molecules such as CCR7, IL7R, and CD62L, are referred to as precursor or progenitor exhausted T cells (T$_{PEX}$)[6–12]. Compared with the terminally exhausted T cells, T$_{PEX}$ possess superior survival capacity and proliferative potential upon release from immunoinhibitory signals. Novel phenotypic markers, including SLAMF6, CXCR5, and BTLA highly expressed in T$_{PEX}$ than in terminally exhausted T cells have been identified[6,11,13]. Conversely, terminally exhausted T cells preferentially express markers such as CD39, CD244, and TIM3[14,15].

In addition to the tumor-reactive T cells, TILs also include non-tumor reactive bystander T cells[14,16–18]. Bystander TILs express diverse phenotypic markers including memory markers such as CCR7, CD27, and CD28, as well as exhaustion markers such as PD1 and TIGIT[14,19]. The T$_{PEX}$ population cannot be easily identified owing to these complex expression patterns within each TIL subset. Considering the superior long-term cell survival and response to the inhibition of immune checkpoint molecules, it is clinically important to distinguish precursor exhausted TILs from the terminally exhausted and bystander TILs with robust phenotypic marker profiles.

In this study, we aimed to identify a surface molecule that characterizes T$_{PEX}$ and demonstrate that CD83 is predominantly expressed in exhausted T cells with an early memory phenotype. CD83 is a member of the immunoglobulin superfamily and highly expressed in mature dendritic cells to function as one of the costimulation and adhesion molecules[20]. Although activated T cells also upregulate CD83[21], its expression dynamics has not been elucidated in detail. We characterize the expression kinetics and functional roles of CD83 in antitumor T cells.

## Results

### Functional heterogeneity of the antitumor T cells in the solid tumor site
To assess the phenotypic and functional properties of the exhausted antitumor T cells in an experimental model, we subcutaneously inoculated NSG mice with A375-mesothelin and treated them with mesothelin-targeting CAR-T cells (Fig. 1a). Tumor-infiltrating CAR-T cells were collected for ex vivo analysis at different time points (at days 14–34). As expected, most of the CAR-T cells on the tumor site showed higher levels of the immunoinhibitory molecule PD1 and reduced expression of the memory marker CCR7 compared with the in vitro cultured T cells (Fig. 1b). However, we detected a small population of PD1$^+$CCR7$^+$ CAR-T cells, which was significantly decreased in the tumor samples collected at later time points (Fig. 1c). To evaluate the proliferative capacity of the tumor-infiltrating CAR-T cells upon antigenic restimulation, CD8$^+$ CAR-T cells were isolated from the tumor and co-cultured with K562-mesothelin following labeling with CFSE. We observed a clear correlation between T-cell division rate and the frequency of CCR7$^+$ cells, confirming T$_{PEX}$ that retain the proliferative capacity are enriched in the CCR7$^+$ CAR-T cell population (Fig. 1d–f).

### CD83 expression marks precursor exhausted T cell population
We then sought to identify robust markers that were preferentially expressed in the T$_{PEX}$. We extracted six surface molecule-encoding genes (BTLA, CCR6, CD81, CD83, CRTAM, and SLAMF6) that were highly expressed in T$_{PEX}$ than in the terminally exhausted T cells as well as conventional memory T cells using publicly available gene expression data (Fig. 2a)[6–8]. These candidate molecules included those previously reported to be differentially expressed between the precursor and terminally exhausted T cells such as SLAMF6 and BTLA[6,13]. We then compared the cell-surface expression of the individual molecules between CCR7$^+$PD1$^+$ (enriched in T$_{PEX}$) and CCR7$^-$PD1$^+$ CAR-T cells as well as PD1$^-$ non-CAR-T cells within the solid tumor (Fig. 2b). Expression of IL7R (upregulated in the memory T cells) was also analyzed as a reference. As shown in Fig. 2c, most of the analyzed markers including IL7R were preferentially expressed in the CCR7$^+$ T-cell population. Although CD81 and SLAMF6 were mostly expressed in all of the T cell subsets, these molecules also showed increased expression levels in CCR7$^+$PD1$^+$ T cells compared with the other populations when compared by mean fluorescence intensity (Supplementary Fig. 1). Among the candidate genes, CD83 expression showed the most selective expression in the CCR7$^+$PD1$^+$ CAR-T cell population compared with the CCR7$^-$PD1$^+$ and PD1$^-$ T cells (Fig. 2c, d). In addition, CD83 was rarely expressed in peripheral blood CD8$^+$ T cells, irrespective of CCR7 positivity (Fig. 2e). We validated these results using four additional tumor samples (Fig. 2f).

To examine the expression of CD83 in tumor-infiltrating CAR-T cells compared with T cells outside the tumor, we administered anti-mesothelin CAR-T cells intratumorally into subcutaneous A375-mesothelin tumors and uncultured T cells from the same donor intravenously (Fig. 3a). When analyzed on day 5 after T-cell injection, CD83 was more upregulated in the intratumoral CCR7$^+$ CAR-T cell population than in CCR7$^-$ CAR-T cells in the tumor and T cells in the spleen (Fig. 3b, c). The expression of CCR7 was selectively observed in the CD83$^+$PD1$^+$ cells but not in the CD83$^-$PD1$^+$ T cell population (Fig. 3d, e). Both CCR7$^+$ and CCR7$^-$ CAR-T cells similarly expressed TIM3 at high levels compared to T cells in the spleen (Fig. 3f). Further phenotypic analysis showed that CD83$^+$ CAR-T cells expressed increased levels of TCF7 and, conversely, decreased levels of granzyme B compared to CD83$^-$ CAR-T cells (Fig. 3g–j). These phenotypic features are consistent with those of previously described precursor exhausted T cells[6,22,23].

We further investigated functional properties of CD83$^+$ CAR-T cells. To obtain enough cell numbers, we exploited an in vitro chronic stimulation protocol that was reported to induce dysfunctional T cells mimicking exhausted T cells (Fig. 3k)[24,25]. Since CD83$^+$ T cells included both CCR7$^+$ and CCR7$^-$ cells, and CD83$^-$ cells were mostly negative for CCR7 as was seen in the tumor model, we isolated three populations (CD83$^+$CCR7$^+$, CD83$^-$CCR7$^+$, and CD83$^-$ cells) and restimulated them in vitro (Fig. 3l). As shown in Fig. 3m and n, CD83$^+$CCR7$^+$ CAR-T cells displayed superior proliferation and IL-2 production compared with the other subsets, which are previously reported features of precursor exhausted T cells[6,8,10]. There was no significant difference in the secretion of IFN-γ among the three populations (Fig. 3o). These results collectively suggest that T cells double positive for CD83 and CCR7 possess functional properties of precursor exhausted T cells.

### CD83 is induced upon T cell activation
We next analyzed the expression kinetics of CD83 in CD8$^+$ T cells. Consistent with the previous studies[21], we confirmed that CD83 expression was rapidly induced in CAR-T cells upon antigenic stimulation and subsequently downregulated (Fig. 4a, b). We then investigated whether

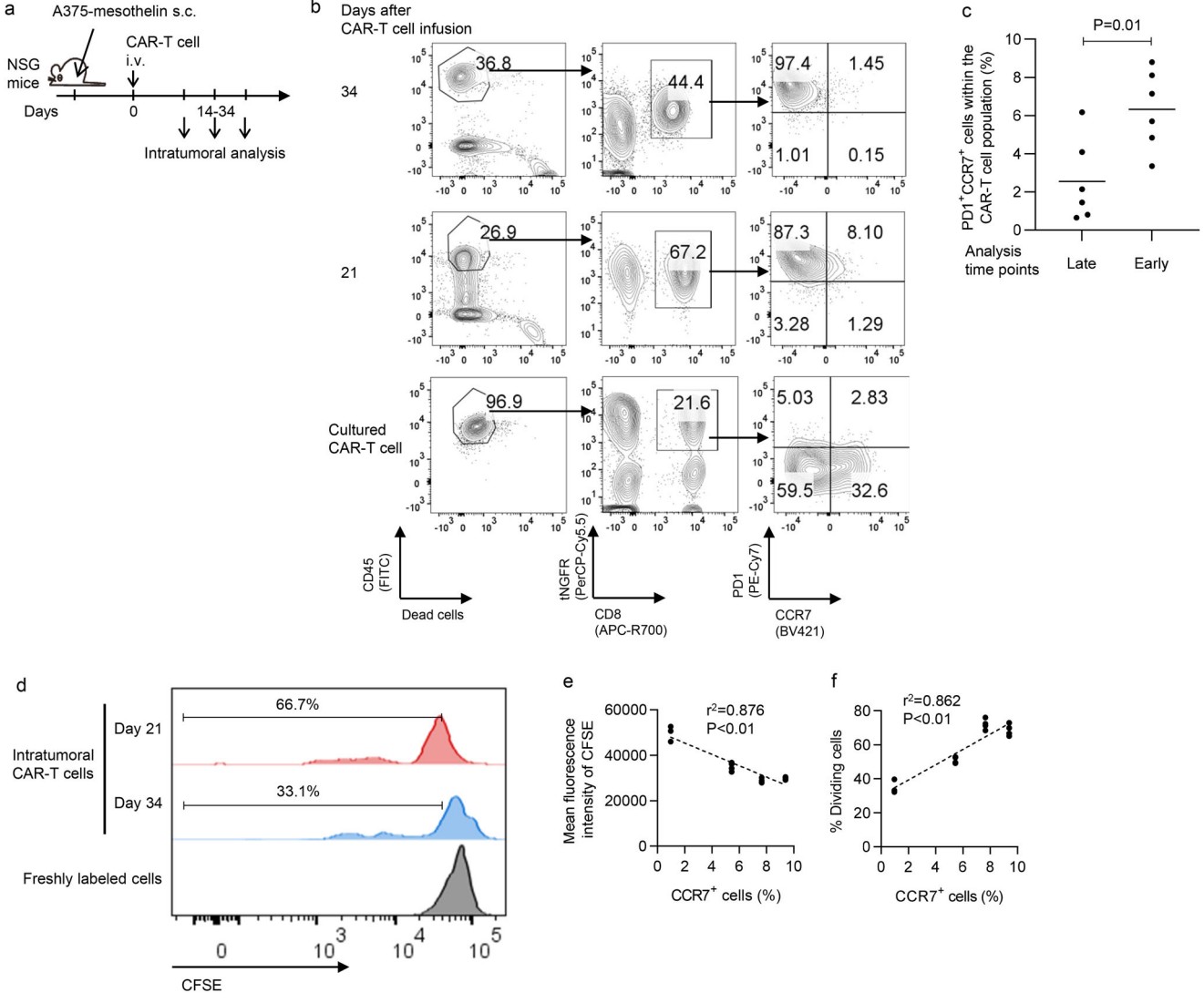

**Fig. 1 Precursor exhausted T cells within the tumor-infiltrating CAR-T cells. a** NSG mice subcutaneously inoculated with the A375-mesothelin were treated using mesothelin-targeting CAR-T cells. Intratumoral CAR-T cells were assessed for phenotypic and functional properties at various time points. **b** Representative flow cytometry plots analyzing PD1 and CCR7 expression. **c** The frequency of PD1+CCR7+ cells in the intratumoral CD8+ CAR-T cells at early (day 14 or 21) or late (day 26 or 34) time points ($n = 6$ mice for each group, unpaired two-tailed $t$-test). Horizontal lines indicate mean values. **d** Intratumoral CAR-T cells isolated on day 21 or 34 were stimulated in vitro by K562-mesothelin following CFSE labeling. Representative plots analyzing CFSE dilution 6 days after restimulation. **e, f** Mean fluorescence intensity of CFSE (**e**) and the frequency of CFSE-diluted dividing cells (**f**) were plotted against the frequency of CCR7+ cells within the CAR-T cell population at the time of T-cell extraction ($n = 3$–4 cultures for each of the 4 different sample, $P$-values and coefficients were calculated by Pearson correlation analysis).

T-cell differentiation status affects CD83 upregulation upon antigen stimulation. Repeated stimulation progressively promoted the terminal T cell differentiation (Fig. 4c, d; Supplementary Fig. 2a). We observed that the terminally differentiated CAR-T cells displayed reduced upregulation of CD83 following repeated antigen encounters (Fig. 4e, f). These results are consistent with the in vivo findings that CCR7+ less-differentiated CAR-T cells exposed to the target antigen showed higher expression levels of CD83 than CCR7− CAR-T cells. Similar results were obtained in the peripheral blood T cells stimulated with anti-CD3 mAb (Supplementary Fig. 2b, c). Ligation with anti-CD3 mAb alone was sufficient to induce CD83 expression, and co-stimulation with CD80 further augmented this upregulation (Fig. 4g, h).

**CD83 overexpression limits effector T cell functions.** We next investigated functional significance of CD83 expression in

activated T cells. T cells were ectopically expressed with CD83 and repeatedly stimulated by K562-OKT3/CD80. Although CD83-engineered T cells initially expanded similarly to control T cells, they exhibited decreased proliferation upon repeated stimulation (Fig. 5a, b). Consistent with these results, CD83-overexpressing T-cell population decreased at later time points (Fig. 5c). We also confirmed that CD83-overexpressing T cells showed slower cell division rates compared with the control (Fig. 5d, e). Similar results were obtained in CAR-T cells upon repeated antigen stimulation (Supplementary Fig. 3a, b). Moreover, ectopic expression of CD83 in CAR-T cells significantly attenuated production of cytolytic effector molecules and CD107a surface expression (Fig. 5f, g). CD83 overexpression did not affect cytokine production by CAR-T cells (Fig. 5h). To investigate the effect of CD83 overexpression on antitumor efficacy of CAR-T cells in vivo, we infused mesothelin-targeting CAR-T cells with or without ectopic

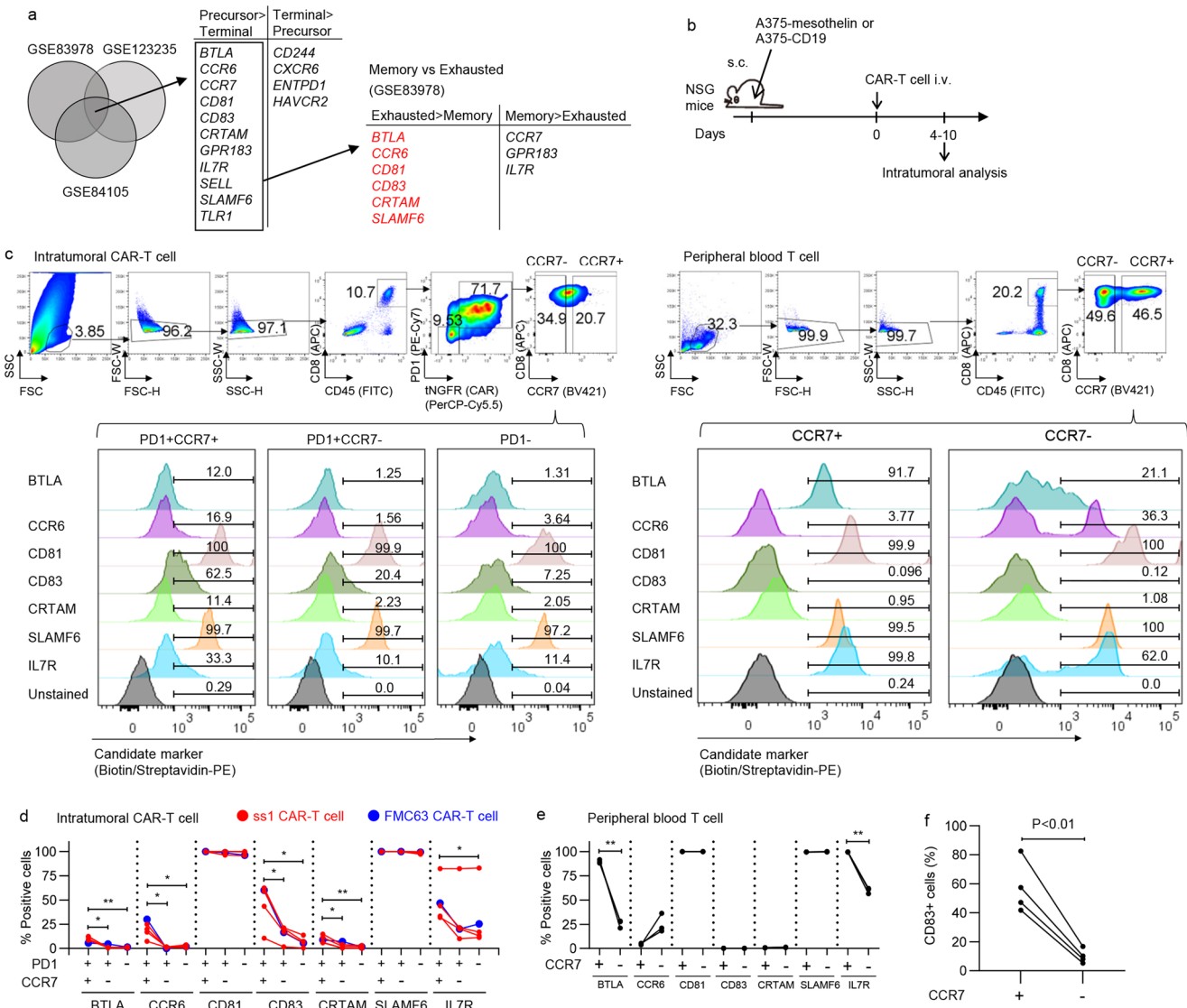

**Fig. 2 CD83 is predominantly expressed in the precursor exhausted T cells. a** Extraction of the candidate surface marker genes with predominant expression in a precursor exhausted T cell population compared with the terminally exhausted or non-exhausted memory T cells. **b–d** NSG mice were subcutaneously inoculated with A375-mesothelin or A375-CD19 and then treated with CAR-T cells. **c** Representative flow cytometry plots analyzing the expression of individual molecules in the tumor-infiltrating T cells and peripheral blood T cells. Gating threshold for each molecule was determined based on the plots of the fluorescence minus one (FMO) control samples. **d, e** The frequency of cells that express the indicated molecules in the CCR7$^+$PD1$^+$, CCR7$^-$PD1$^+$, and PD1$^-$ T cell population within the tumor (**d**), ($n = 5$ samples from different mice, repeated measures one-way ANOVA with multiple comparison test) and in the CCR7$^{+/-}$ peripheral blood T cells (**e**), ($n = 3$ different samples, two-tailed paired $t$-test). *$P < 0.05$, **$P < 0.01$. **f** CD83 expression in the CCR7$^{+/-}$ intratumoral CAR-T cells was analyzed in the additional four mice using the A375-mesothelin model ($n = 4$ samples from different mice, paired two-tailed $t$-test).

expression of CD83 into NSG mice that were subcutaneously inoculated with the mesothelin-expressing pancreatic cancer cell line AsPC-1 (Fig. 5i). We also transduced CAR-T cells with the luciferase gene to monitor their persistence. CD83-overexpressing CAR-T cells initially showed less efficient control of tumor growth, which is consistent with the in vitro data that ectopic expression of CD83 attenuated the production of cytolytic molecules (Fig. 5j, k). When monitored by the luciferase activity, both control and CD83-overexpressing CAR-T cells progressively accumulated at the tumor site and eventually controlled tumor growth in most of the treated mice (Fig. 5l; Supplementary Fig. 4). These results suggest that CD83 is not merely a surface marker characterizing T$_{PEX}$ but has functional significance in some of the effector T cell functions.

**CD83 is predominantly expressed in naturally occurring TILs with a precursor phenotype**. We investigated CD83 expression of tumor-infiltrating T cells in immunocompetent mouse tumor models. We used a Colon-26 mouse tumor model, in which infiltration of the exhausted CD8$^+$ T cells was observed in the subcutaneously inoculated tumor tissue (Fig. 6a)[26]. As was observed in the human T-cell studies, CD83 and PD1 were upregulated in a subset of the CD8$^+$ TIL population, which was not seen in the T cells from the spleen (Fig. 6b, c). A majority of the intratumoral CD8$^+$ T cells have differentiated into an effector memory phenotype (CD44$^+$CD62L$^-$: T$_{EM}$), in contrast to the spleen T cells possessing CD44$^-$CD62L$^+$ (naïve (T$_N$)) and stem-cell memory (T$_{SCM}$) population) and CD44$^+$CD62L$^+$ (central memory (T$_{CM}$)) population) cells. CD83 expression was predominantly detected in the TIL population with a T$_{CM}$ phenotype

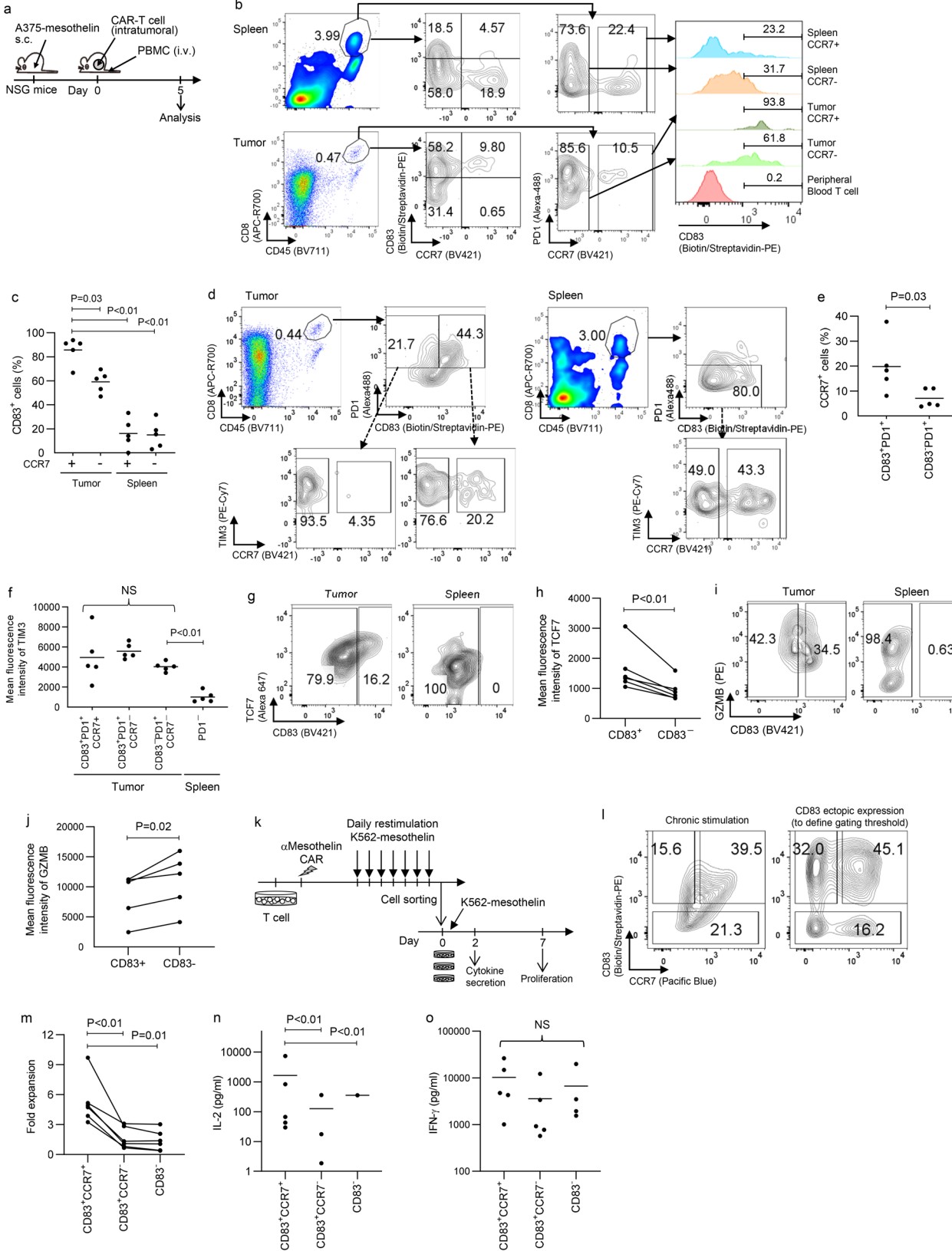

(Fig. 6d). These results suggest that naïve T cells do not express CD83, and differentiated T cells with a T_EM phenotype are attenuated in the upregulation of CD83. In contrast, PD1 expression was the most prominent in the T_EM population. We observed similar findings in the B16 melanoma tumor model (Supplementary Fig. 5a–c). To further confirm these results, we reanalyzed publicly available single-cell gene expression data of mouse CD8$^+$ T cells infiltrating B16 melanoma tumor tissue[27]. Among the six clusters identified by unsupervised analysis (Fig. 6e), clusters 2 and 4 included T cells that expressed memory-related genes (*Ccr7*, *Tcf7*, *Sell*, and *Il7r*), whereas exhaustion-related genes (*Pdcd1*, *Lag3*, *Havcr2*, *Tox*, and *Nr4a1*) were mainly

**Fig. 3 CD83-expressing tumor-infiltrating CAR-T cells display phenotypic and functional attributes of precursor exhausted T cells. a–i** NSG mice subcutaneously inoculated with the A375-mesothelin were infused with mesothelin-targeting CAR-T cells intratumorally and peripheral blood T cells from the same donor intravenously. T cells within the tumor and spleen were analyzed 5 days after infusion. **b, c** The expression of CD83 was analyzed in $CCR7^{+/-}$ T cells within the tumor and spleen. Representative flow cytometry plots (**b**) and the frequency of $CD83^+$ cells in the $CCR7^{+/-}CD8^+$ T cell populations are shown (**c**), ($n = 5$ mice, repeated measures one-way ANOVA with multiple comparison test). **d–f** The expression of CCR7 and TIM3 was compared between $CD83^{+/-}PD1^+CD8^+$ CAR-T cells within the tumor. The data shown are representative flow cytometry plots (**d**), the frequency of $CCR7^+$ cells in $CD83^{+/-}PD1^+$ CAR-T cells in the tumor (**e**), ($n = 5$ mice, paired two-tailed $t$-test), and mean fluorescence intensity of TIM3 in the indicated T cell populations (**f**), ($n = 5$ mice, repeated measures one-way ANOVA with multiple comparison test). The data in **c**, **e**, and **f** are derived from the same mice. **g–j** Expression levels of TCF7 and granzyme B of intratumoral CAR-T cells were analyzed by intracellular flow cytometry. The data shown are representative flow cytometry plots (**g**, **i**) and the mean fluorescence intensity of TCF7 (**h**) and granzyme B (**j**) in the $CD83^+$ and $CD83^-$ CAR-T cell populations. ($n = 6$ or 5 mice, paired two-tailed $t$-test). **k–o** Mesothelin-targeting CAR-T cells were daily stimulated with K562-mesothelin for 7 days, and $CCR7^+CD83^+$, $CCR7^-CD83^+$, and $CD83^-$ $CD8^+$ CAR-T cells were purified by flow cytometry (**l**), (CD83-transduced cultured T cells were analyzed using the same panel to determine gating threshold). The isolated T cells were then restimulated by K562-mesothelin to analyze fold expansion (**m**), ($n = 6$ different samples) and the secretion of IL-2 (**n**), ($n = 5$ different samples; two samples from $CD83^+CCR7^-$ and four samples from $CD83^-$ cells were under detection limit) and IFN-γ (**o**), ($n = 5$ different samples; one sample from $CD83^-$ cells was under detection limit). In (**m–o**), statistical significance was tested by repeated measures one-way ANOVA with multiple comparison test. For (**n**) and (**o**), log-transformed values were used for calculation. Horizontal lines indicate mean values. NS not significant.

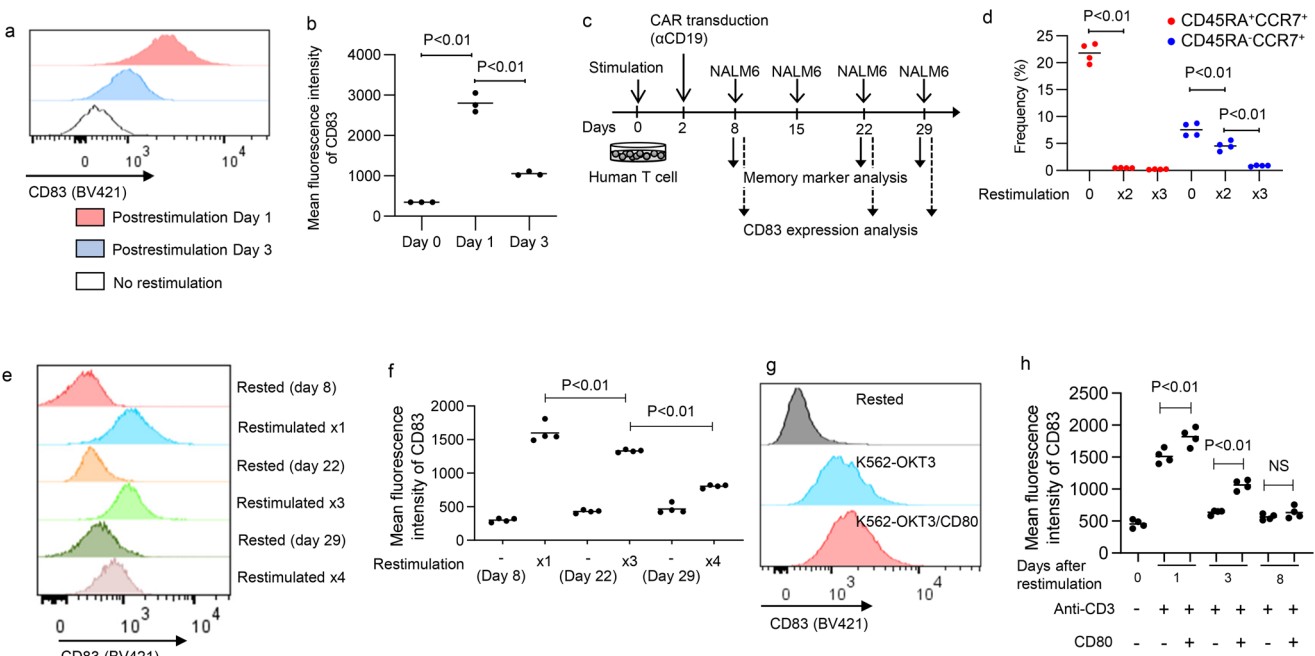

**Fig. 4 CD83 upregulation upon antigen stimulation is attenuated along with terminal T cell differentiation. a, b** CD19-targeting CAR-T cells were stimulated by NALM6 in vitro and analyzed for CD83 expression at the indicated time points. **a** Representative flow cytometry plots in the $CD8^+$ CAR-T cell population. **b** The mean fluorescence intensity (MFI) of CD83 is shown ($n = 3$ independent cultures, ordinary one-way ANOVA with multiple comparison test). **c** CD19-targeting CAR-T cells were repeatedly stimulated by NALM6. **d** The frequency of $CD45RA^{+/-}CCR7^+$ memory T-cell population was analyzed at the indicated time points ($n = 4$ independent cultures, ordinary one-way ANOVA with multiple comparison test). **e, f** CD83 expression was analyzed in the $CD8^+$ CAR-T cell population upon repeated NALM6 stimulation. The data shown are representative flow cytometry plots (**e**) and the calculated CD83 MFI (**f**), ($n = 4$ independent cultures, ordinary one-way ANOVA with multiple comparison test). **g, h** Cultured T cells were restimulated by K562-OKT3 or K562-OKT3/CD80 and analyzed for CD83 expression. Representative flow cytometry plots (**g**) and the CD83 MFI at days 1, 3, and 8 after restimulation are shown (**h**), ($n = 4$ independent cultures, unpaired two-tailed $t$-test at each time point). Horizontal lines indicate mean values.

enriched in clusters 0, 2, 3, and 5 (Fig. 6f; Supplementary Fig. 6). As only T cells within cluster 2 showed dual expression of memory-related, as well as exhaustion-related genes, this cluster was considered to represent a population of $T_{PEX}$. Gene set enrichment analysis among the clusters verified that the $T_{PEX}$-associated gene set was significantly enriched in the cluster 2 compared with the other clusters (Fig. 6g). CD83 expression was selectively observed in T cells within cluster 2, thereby reinforcing our finding that its expression is skewed in a $T_{PEX}$ population.

Finally, we examined the distribution of CD83 expression in endogenous human tumor-infiltrating T cells. Resected tumor samples from patients with ovarian, cervical, and endometrial cancer were analyzed for CD83 expression. As shown in Fig. 7a, b, CD83 was significantly upregulated in the $CCR7^+PD1^+$ T cell subset compared with that in the $PD1^-$ or $CCR7^-PD1^+$ T cells in the $CD8^+$ T cell populations. We also analyzed the publicly available single-cell RNA-seq dataset of TIL samples derived from various cancer types: GSE156728 (pan-cancer)[28], GSE98638 (hepatocellular carcinoma)[29], GSE115978 (melanoma)[30], GSE120575 (melanoma)[31], and GSE190202 (breast cancer)[32]. Overall, the frequency of CD83-expressing cells was significantly higher in the $CD8^+$ T cells that expressed both PD1 and CCR7 than in the $CCR7^-PD1^+$ or $PD1^-$ T cells, which is consistent with the above results (Fig. 7c). In contrast, *HAVCR2*

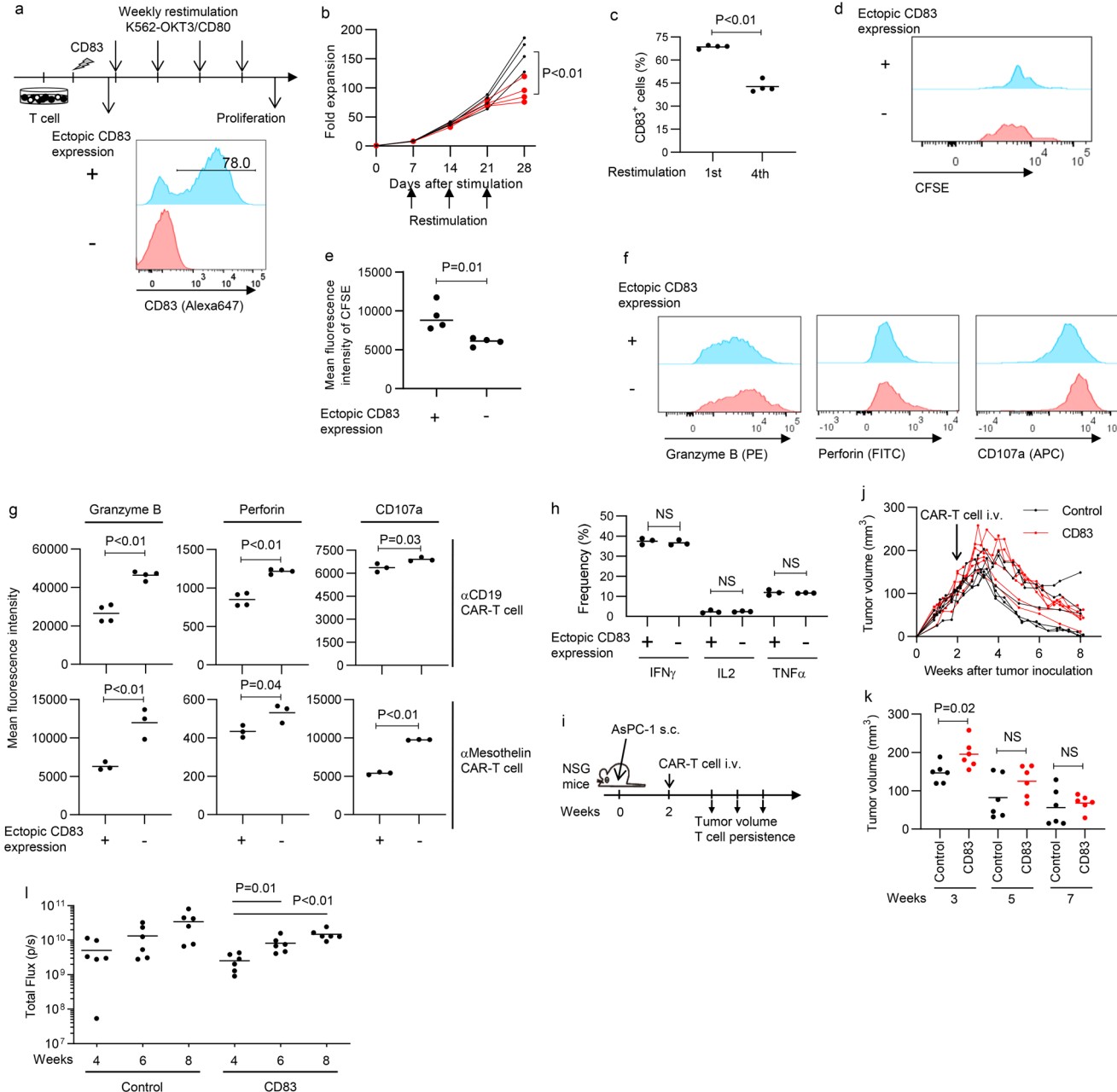

**Fig. 5 CD83 expression negatively affects CD8+ T-cell effector functions. a–c** Human T cells were retrovirally transduced with CD83 and weekly restimulated by K562-mOKT3/CD80. **b** Fold expansion of T cells. Arrows indicate T cell stimulation by K562-mOKT3/CD80 ($n = 4$ independent cultures, representative data of two experiments). **c** The frequency of CD83-transduced cells within the CD8+ T cell population was analyzed with flow cytometry after the 1st or 4th stimulation ($n = 4$ independent cultures). **d, e** Control or CD83-overexpressing T cells were analyzed for CFSE dilution upon the 4th restimulation. Representative flow cytometry plots (**d**) and the mean fluorescence intensity of CFSE 5 days after stimulation are shown (**e**), ($n = 4$ independent cultures). **f, g** Control or CD83-overexpressing anti-CD19 or anti-mesothelin CAR-T cells were analyzed for production of granzyme B and perforin and CD107a expression upon restimulation by NALM6 or K562-mesothelin. Representative flow cytometry plots for anti-mesothelin CAR-T cells (**f**) and mean fluorescence intensity are shown (**g**), ($n = 3$-4 independent cultures). **h** CD19-targeting CAR-T cells with or without ectopic expression of CD83 were analyzed for cytokine production upon restimulation by NALM6 ($n = 3$ independent cultures). **i** NSG mice subcutaneously transplanted with the mesothelin+ pancreatic cancer cell line AsPC-1 were treated by control or CD83-overexpressing CAR-T cells against mesothelin. CAR-T cells were cotransduced with the luciferase gene to monitor in vivo persistence. **j, k** Tumor volume was longitudinally monitored ($n = 6$ mice for each). **l** Total flux of the luciferase activity was analyzed by in vivo bioluminescent imaging ($n = 6$ mice, repeated measures one-way ANOVA with multiple comparison test for the log-transformed values). The data presented in (**j–l**) are a composite of three independent experiments. In (**b, c, e, g, h,** and **k**), statistical significance was analyzed by unpaired two-tailed $t$-test. NS not significant.

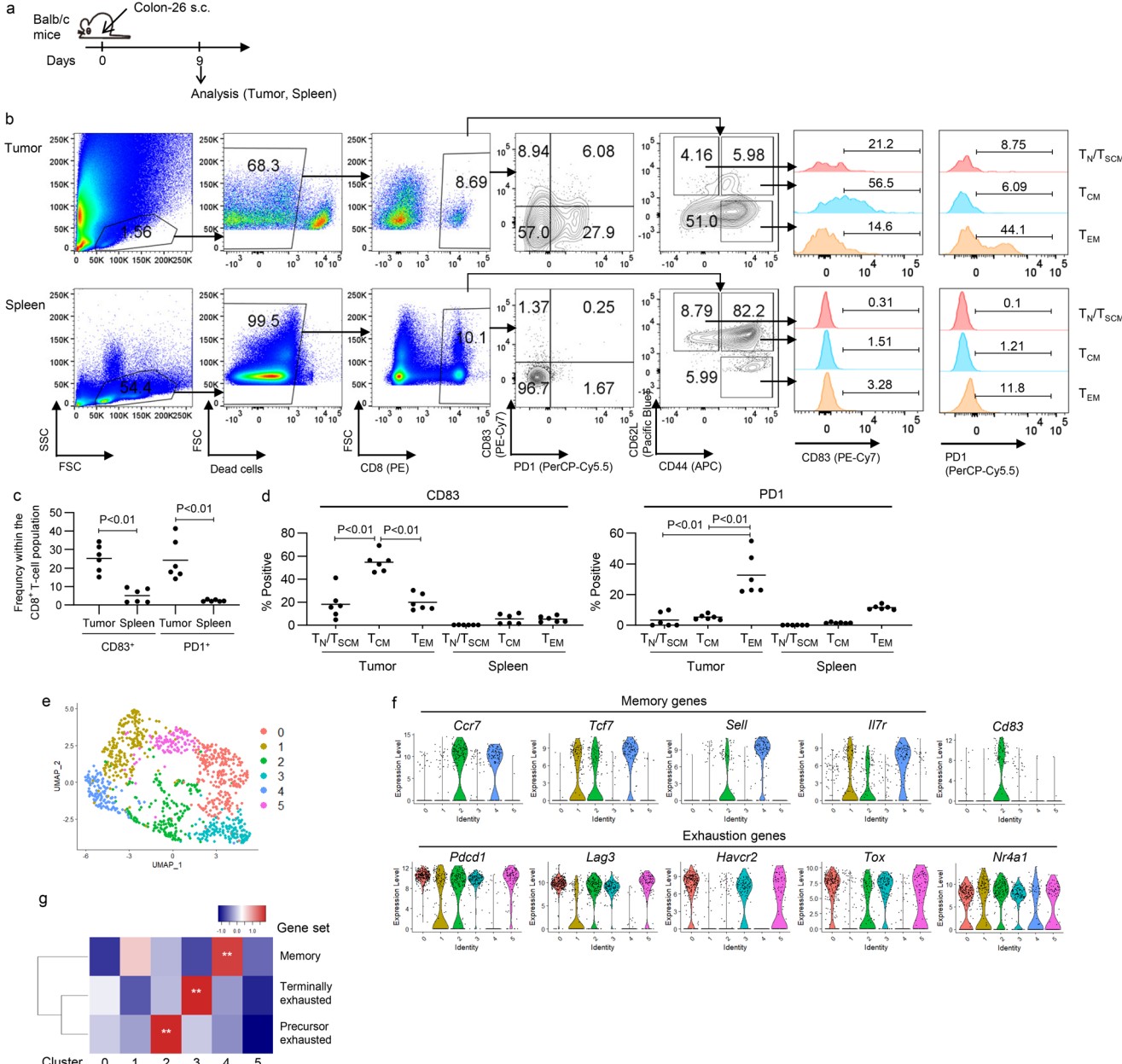

**Fig. 6 CD83 is preferentially expressed in naturally occurring tumor-infiltrating T cells with a central memory phenotype. a** BALB/c mice were subcutaneously injected with the Colon-26 cell line. Tumor-infiltrating T cells (TILs) were analyzed for CD83, PD1, and memory markers (CD44 and CD62L) on day 9. **b, c** Representative flow cytometry plots (**b**) and the frequency of CD83$^+$ and PD1$^+$ T cells within the subcutaneous tumor and spleen (**c**), ($n = 6$ samples from different mice, unpaired two-tailed $t$-test for each). **d** CD83 and PD1 expression were analyzed for the indicated memory T-cell subsets ($n = 6$ samples from different mice, repeated measures one-way ANOVA with multiple comparison test). **e** Single-cell RNA-sequencing analysis for CD8$^+$ T cells infiltrating the B16 melanoma cells from the publicly available data were clustered and visualized by UMAP. **f** Violin plots showing the expression of the indicated genes within each cluster. **g** Heatmap of the gene set enrichment analysis for the genes associated with precursor exhausted, terminally exhausted, and memory T cells among the six clusters identified in single-cell RNA-seq analysis. **P < 0.01.

(TIM3)-expressing T cells were significantly more abundant in the CCR7$^-$PD1$^+$ population than the other subsets. We also explored if the expression levels of CD83 are associated with prognosis of the patients treated with immune checkpoint inhibitors (ICI) using one of the above scRNA-seq datasets (GSE120575). When we counted the total number of CD83$^+$CCR7$^+$ cells in each of the four groups (pre- or posttreatment melanoma samples from responders or non-responders to ICI), the proportion of CD83$^+$CCR7$^+$ cells within the CD8$^+$ T cell population was higher in the responder group (43 of 1005 cells, 4.3%) than in the non-responder group (26 of

1587 cells, 1.6%) at baseline, which significantly decreased after ICI treatment (25 of 1082 cells, 2.3%) (Fig. 7d). However, the frequency of CD83$^+$CCR7$^+$ cells calculated for individual patients' samples was not significantly different between responders and non-responders to ICI (Fig. 7e). These results might be due to the insufficient number of patients or because CD83 has poor predictive power when used as a single marker. We further analyzed multiple bulk RNA-seq data with prognostic information[33–36]. Although CD83 expression levels were related to therapeutic response to ICI in some of the data, its high expression was not significantly associated with survival of the

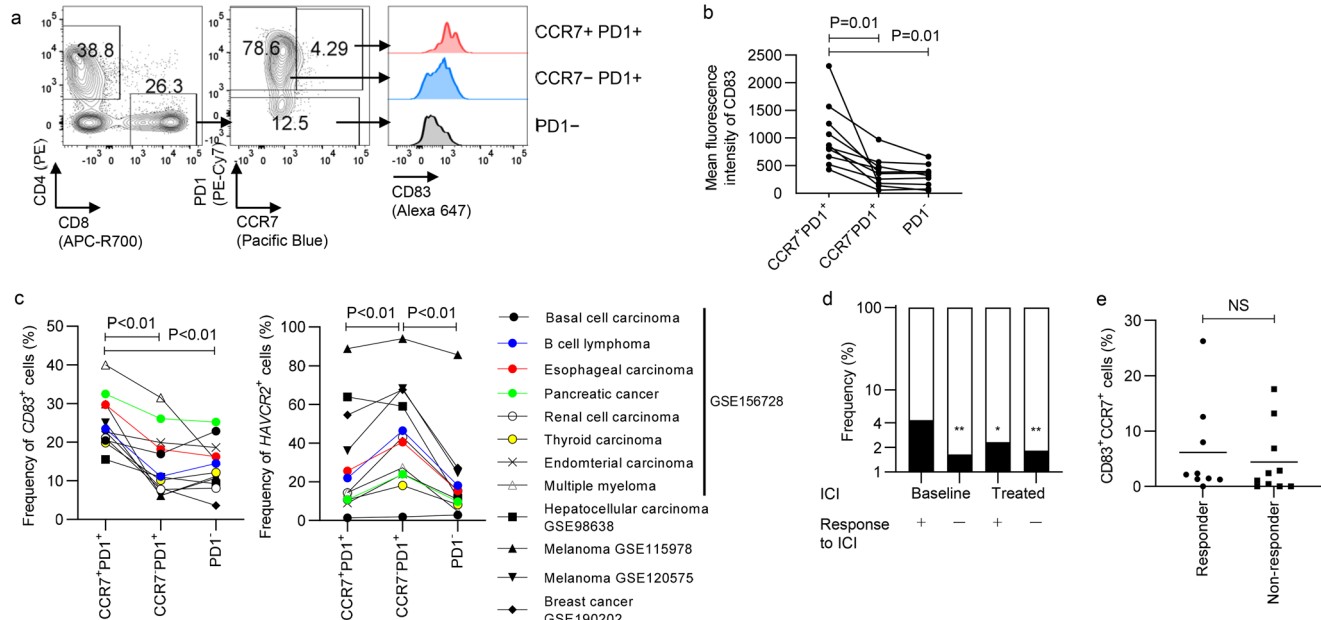

**Fig. 7 CD83 is expression is skewed in the CCR7+PD1+ T-cell population within the tumor tissue. a, b** CD83 expression was analyzed in TIL samples from patients with gynecologic malignancies. The data shown are representative flow cytometry plots (**a**) and the mean fluorescence intensity of CD83 in the CD8+ T-cell populations (**b**), ($n = 10$ different samples, repeated measures one-way ANOVA with multiple comparison test). **c** Single-cell RNA-sequencing data of TIL from various cancer types were analyzed for the proportion of CD83+ cells within the CCR7+PD1+, CCR7-PD1+, and PD1− CD8+ T cell populations ($n = 12$ datasets, repeated measures one-way ANOVA with multiple comparison test). **d** In the GSE120575 data, the proportion of CD83+CCR7+ cells within the total CD8+ T cells were compared among the samples collected before or after treatment with immune checkpoint inhibitors in the responder or non-responder patients ($n = 1005, 1587, 1082$, and $3067$ cells, *$P < 0.05$, **$P < 0.01$, Fisher's exact test with multiple testing correction). **e** The frequency of CD83+CCR7+ cells was calculated in the individual patients' samples with or without response to immune checkpoint inhibitors using the GSE120575 dataset ($n = 9$ or $10$ patients, unpaired two-tailed $t$-test; NS not significant).

patients, suggesting that CD83 expression alone cannot predict overall prognosis of the patients treated with ICI (Supplementary Fig. 7a–e).

## Discussion

Phenotypic and qualitative diversity of TILs has been elucidated in multiple studies. From a therapeutic standpoint, the frequency of $T_{PEX}$, which are exhausted but possess an early memory phenotype, is associated with effective cancer immunotherapy. $T_{PEX}$ but not terminally exhausted TILs can regain effector functions upon immune checkpoint blockade[6]. The $T_{PEX}$ population is also associated with the efficacy of adoptive immunotherapy using the in vitro expanded TILs[37,38]. In addition to exhausted tumor-reactive T cells, endogenous TILs also include bystander T cells that recognize non-tumor antigens. Since both $T_{PEX}$ and bystander T cells share the molecular signature of memory T cells, robust phenotypic markers are required to discriminate between these subsets[14,39,40]. We demonstrated that CD83 expression in T cells is induced only upon antigen stimulation. We have also shown that the activation-mediated CD83 upregulation is attenuated in terminally differentiated T cells, and thereby its expression is selectively detected in $T_{PEX}$. Considering the importance of the $T_{PEX}$ population to reinvigorate antitumor immunity, analysis of CD83 expression in TIL samples may be useful as a biomarker to predict responsiveness to immunotherapeutic modalities in combination with other markers. Moreover, isolation of CD83+ TILs may facilitate cloning of cancer neoantigen-specific TCR with optimal affinity. T cells with high-affinity tumor-reactive TCR can be efficiently enriched using markers expressed in terminally exhausted T cells[17,41–43]. In virus infection, while high-affinity antiviral T cells play a dominant role in viral clearance at the acute phase, T cells with low- to intermediate-affinity TCR persist in the

chronic phase[44–46]. Although the difference of tumor-reactive TCR repertoire between precursor exhausted and terminally exhausted T cells has not been investigated enough, the relative–abundance of $T_{PEX}$ is essential for durable clinical response in TIL therapy, suggesting that the $T_{PEX}$ population contains optimal-avidity antitumor T cells for durable response[38]. In addition to CD83, we have also found several other markers that are predominantly upregulated in $T_{PEX}$, including BTLA, CCR6, and CD81. Several other molecules have recently been reported to have predominant expression in a subset of exhausted T cells, which include CXCL13[28,47], LAYN[48], and CD69[38]. Combinatorial use of these markers may further enhance detection of tumor-reactive T cells with superior survival potential. SLAMF6, one of the previously identified $T_{PEX}$ markers in mouse tumor and virus infection models, did not show a marked difference between CCR7+ and CCR7− exhausted T cells[6,8]. One possibility is that there is a different transcriptional regulation between human and mouse T cells. In addition, our analysis was performed using genetically engineered CAR-T cell treatment models, which have substantially different conditions compared with endogenous antitumor T cells within the tumor, including the target antigen density and avidity of the T cells. Considering that SLAMF6 is quickly upregulated after T cell activation[49], its expression levels may significantly be affected by these parameters. Similarly, although previous studies demonstrated a predominant expression of TIM3 in terminally exhausted T cells[6–12], we did not observe a prominent difference in the expression levels of TIM3 between CD83+/− CAR-T cells in our models. Nevertheless, CD83+CCR7+ intratumoral T cells possessed at least a part of functional features of $T_{PEX}$ such as superior proliferation and IL-2 production.

While CD83 is associated with the development and function of regulatory T cells[50,51], its role in CD8+ T cells has not been explored in detail. Although the specific ligand for CD83 has not

been confirmed, recent studies have reported its homotypic interaction, suggesting that activated T cells may interact with each other through upregulated CD83[52]. Intriguingly, ectopic expression of CD83 negatively affected T-cell proliferation and cytotoxic activity upon antigen encounter. Likewise, SLAMF6 and BTLA, which have recently been reported to be expressed in $T_{PEX}$, have an inhibitory effect on effector T cell functions[12,13]. These molecules may play a role in suppressing the excessive activation and subsequent terminal differentiation of antitumor T cells. CD83 has an essential role in the development and functions of regulatory T cells[51,53]. On the other hand, previous studies investigating the effect of CD83 ligation on conventional T cells have been controversial. While membranous CD83 expression in dendritic cells promotes T-cell expansion and effector functions[54,55], a soluble form of CD83 inhibits T cell proliferation[21,56]. These apparently discrepant results may suggest that the optimal strength of CD83 exists for efficient T cell expansion. For example, although MAPK signaling is necessary for T cell proliferation and effector functions, excessive signaling compromises long-term persistence of endogenous as well as CAR-engineered T cells[57,58]. Further investigations are required to elucidate downstream signaling of CD83 ligation and the effect on T cell functions.

In summary, the present study elucidates CD83 as a surface molecule to identify $T_{PEX}$ within the heterogeneous TILs. These findings can be applicable to evaluate the quality and therapeutic potential of TILs used for successful cancer immunotherapy.

## Methods

**Cell lines**. The erythroleukemia cell line K562 and the mouse melanoma cell line B16 were purchased from the Japanese Collection of Research Bioresources cell bank (Osaka, Japan). The CD19+ B-cell leukemia cell line NALM6 and the mouse colon carcinoma cell line Colon-26 were obtained from the Cell Resource Center for Biomedical Research, Tohoku University (Sendai, Japan). The A375 melanoma, PG13 retroviral packaging cell line, and AsPC1 pancreatic cancer cell line were obtained from the American Type Culture Collection (Manassas, VA, USA). The Plat-A and Plat-E packaging cell lines were kindly provided by Dr. T. Kitamura (University of Tokyo, Tokyo, Japan). K562, NALM6, Colon 26, AsPC1, and their derivatives were cultured in RPMI-1640 (Nacalai Tesque, Japan) containing 10% fetal bovine serum (FBS) (Nichirei Biosciences, Tokyo, Japan). A375 and PG13 cells were cultured in DMEM (Nacalai Tesque, Kyoto, Japan) containing 10% FBS. B16 cells were cultured in MEMα (Nacalai Tesque) supplemented with 10% FBS. A375 cells were transduced with CD19 and mesothelin to generate A375-CD19 and A375-mesothelin, respectively.

**In vitro culture of human T cells**. Healthy donor-derived peripheral blood mononuclear cells (PBMCs) were purchased from Cellular Technology Limited (Cleveland, OH, USA). PBMCs were stimulated with mitomycin C-pretreated K562 cells that express the anti-CD3 single-chain variable fragment (scFV) (derived from the clone OKT3 with modified amino acid sequence) linked to the transmembrane and cytoplasmic domains of CD8α and CD80 (K562-mOKT3/CD80) at an effector to target (E:T) ratio of 7:1[59,60]. The stimulated T cells were then cultured with recombinant IL-2 (100 IU/mL, PeproTech, Rocky Hill, NJ, USA). Retroviral transduction of a CAR gene was performed on day 2 following the initial T-cell stimulation using RetroNectin (Takara Bio, Kusatsu, Japan). We generated stable PG13 virus-packaging cells by treating them with Plat-E-derived retroviruses. These PG13-derived retroviruses were used for T cell infection. The CD19-targeting CAR gene contains the clone FMC63-derived scFV and intracellular domains of CD28 and CD3ζ (FMC63-28z). The mesothelin-targeting CAR gene was constructed by linking the clone ss1-derived scFV with the 28z signaling domain (ss1-28z). Both of the CAR-encoding genes were linked to a truncated form of nerve growth factor receptor (tNGFR) using a Furin-SGSG-P2A sequence to differentiate CAR-transduced T cells and inserted into the pMX retroviral plasmid (provided by Dr. T. Kitamura)[60]. We restimulated CD19-targeting and mesothelin-targeting CAR-T cells using NALM6 (E:T ratio of 1:1) and K562 cells transduced with mesothelin (K562-mesothelin; E:T ratio of 5:1), respectively. The CD83 cDNA was also inserted into the pMX retroviral plasmid.

**Flow cytometry analysis**. Flow cytometry analysis was performed using BD LSRFortessa cell analyzer (BD Biosciences). Antibodies used for the analysis are listed in online Supplementary Table 1. When indicated, dead cells were discriminated using the LIVE/DEAD Fixable Near-IR Dead Cell Stain Kit (Thermo Fisher Scientific). To estimate the cell division rate, T cells were analyzed for the

dilution of carboxyfluorescein diacetate succinimidyl ester (CFSE, Thermo Fisher Scientific) labeled before incubation. The CD83-high/low CAR-T cell population was purified by flow cytometry sorting using FACSAriaIII (BD Biosciences). To analyze granzyme B and perforin production in vitro, CAR-T cells were stimulated by the target antigen for 24 h. The T cells were then fixed and permeabilized using a Cyto-Fast Fix/Perm kit (BioLegend), followed by intracellular staining. Granzyme B expression in intratumoral CAR-T cells was directly analyzed without stimulation. The surface expression of CD107a was analyzed 3 h after antigen stimulation. For cytokine production analysis, anti-CD19 CAR-T cells were restimulated by NALM6 for 2 h, added to Brefeldin A (BioLegend) and further incubated for 4 h. The cells were fixed and permeabilized with a Cyto-Fast Fix/Perm kit (BioLegend) according to the manufacturer's instructions and subjected to intracellular staining. For intracellular flow cytometry analysis of TCF7 expression, T cells were fixed in 4% paraformaldehyde at room temperature for 10 min, permeabilized with ice-cold methanol for 20 min, and stained by anti-TCF7 antibody.

**Quantification of cytokine secretion**. Cytokine production by CAR-T cells was quantified by ELISA according to the manufacturer's instructions. Human IL-2 DuoSet ELISA (R&D Systems, Minneapolis, MN, USA) and Human IFN-gamma DuoSet ELISA (R&D Systems) were used to measure the concentration of IL-2 and IFN-γ, respectively. Mesothelin-targeting CAR-T cells were restimulated by K562-mesothelin, and the culture supernatants were collected 48 h after stimulation. The concentration of samples under the detection limit was regarded as zero for statistical analysis.

**Mouse experiments**. In adoptive immunotherapy models using human CAR-T cells, 4- to 10-week-old male NSG mice (The Jackson Laboratories) were subcutaneously inoculated with one million cells of A375-mesothelin or A375-CD19. The mice underwent intravenous or intratumoral injection of 1–3 million CAR-T cells and were euthanized at days 4–34 for subcutaneous tumor analysis. To evaluate the phenotypes of naturally occurring mouse TILs, 4- to 10-week-old female BALB/c and C57BL/6 mice (The Jackson Laboratories) were subcutaneously injected with Colon-26 and B16 tumor cells, respectively. Mice were euthanized at the indicated time points, and tumor-infiltrating endogenous T cells and spleen T cells were extracted for analysis.

To compare antitumor efficacy between control and CD83-overexpressing CAR-T cells, NSG mice were subcutaneously inoculated with $1.5 \times 10^6$ AsPC1 cells premixed with Matrigel (Corning Inc, Corning, NY) and then treated by mesothelin-targeting CAR-T cells with or without ectopic expression of CD83 at week 2. T cells were transduced with the luciferase gene to quantify their accumulation by the in vivo imaging system (IVIS Lumina II, Perkin Elmer, Waltham, MA). No data were excluded throughout the studies. The investigators were not blinded to group allocation during data collection or analysis.

**RNA-sequencing analysis**. We extracted genes upregulated in the $T_{PEX}$, compared with terminally exhausted T cells using publicly available RNA-sequencing data (GSE83978[8], GSE84105[7], and GSE123235[6]) based on the following criteria: fold discovery rate (FDR) < 0.05 and fold change >4. We further selected genes with elevated expression in $T_{PEX}$ compared with memory T cells (FDR < 0.05 and fold change >2). Genes with lower expression levels in human T cells (average count-per-million values <1) were excluded using our previously performed RNA-sequencing data[60]. Genes encoding the membrane proteins were retrieved from the Human Protein Atlas (https://www.proteinatlas.org/).

RNA-seq data from ERP105482[33], GSE78220[34], GSE126044[35], and GSE135222[36] were used to analyze the association of CD83 expression levels with the prognosis of patients. Patients were stratified into a CD83$^{high}$ or CD83$^{low}$ group using the median CD83 expression values in each cohort as a cut-off point.

**Reanalysis of single-cell RNA-sequencing data**. We analyzed single-cell gene expression data of the mouse CD8+ TILs in the B16 melanoma model (GSE86039)[27] using the Seurat package. Log2-transformed transcripts per million data were used for the analysis. Cells were clustered using the FindClusters function based on the principal component analysis of the 2000 most variably expressed genes and then visualized using the Uniform Manifold Approximation and Projection (UMAP). Violin plots for individual genes were generated using the VlnPlot function. Gene set enrichment analysis among the identified clusters was performed using the function "wmw_gsea" in the singleseqgset R package (version 0.1.2.9000). *P*-values were adjusted with the false discovery rate. Gene sets associated with memory, precursor exhausted, and terminally exhausted T cells were prepared by selecting genes with more than 4-fold increased expression than the other populations in the GSE83978 data.

Single-cell RNA-seq data of TIL (GSE156728[28], GSE98638[29], GSE115978[30], GSE120575[31], and GSE190202[32]) were analyzed using the raw or normalized count files retrieved from the gene expression omnibus (GEO) database. Non-zero read counts were defined as positive expression. For the data from GSE98638, GSE115978, and GSE120575, CD8+ T cells were selected based on the expression of *CD8A* (transcripts per million [TPM] values >3).

**TIL analysis**. Tumor specimens obtained from patients with ovarian, endometrial, and cervical cancer were dissociated into single-cell suspensions using the gentle-MACS Dissociator (Miltenyi Biotech). Pathological diagnosis of the resected tumor tissues is summarized in online Supplementary Table 2.

**Statistics and reproducibility**. The data reproducibility was confirmed by at least three experiments. Individual data points were shown as dots in the graph. The statistical significance of differences between two groups was evaluated using a two-tailed paired or unpaired t-test. Comparisons among three or more groups were performed using ordinary or repeated measures one-way ANOVA with multiple comparison test. When indicated, the log-transformed values were used for comparison. We added 1 to the concentration values of cytokines before log transformation to account for zero values (Fig. 3n, o). Correlation between CFSE dilution and the frequency of CCR7+ cells (Fig. 1e, f) was estimated by Pearson correlation analysis. The frequency of CD83+ or CD83+CCR7+ cells was compared among the groups by Fisher's exact test with multiple testing correction (Fig. 7d). We used a P-value of 0.05 as a threshold to determine statistical significance. All statistical analyses were performed using GraphPad Prism 9 software or R 4.0.5. No statistical method was used to determine the sample size.

**Study approval**. This study was performed in accordance with the Declaration of Helsinki and was approved by the Research Ethics Board of the Aichi Cancer Center, Nagoya, Japan (Approval No. 2020-2-14). Written informed consent was obtained from all the patients who provided TIL samples. All animal experiments were approved by the Animal Care and Use Committee of the Aichi Cancer Center Research Institute, Nagoya, Japan (approval number: R2-8(3)-A).

**Reporting summary**. Further information on research design is available in the Nature Portfolio Reporting Summary linked to this article.

## Data availability

The raw data used in this study are provided in Supplementary Data 1. Other data or information related to this study are available from the corresponding author (Y. K.) upon reasonable request.

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

## Acknowledgements

This work was supported by the Outstanding Young Immunology Researcher Award (Japanese Society for Immunology and Nippon Becton Dickinson Company); JST FOREST Program (Grant Number JPMJFR2060, Japan); JSPS KAKENHI Grant Number 20H03543 (YK); Aichi Cancer Center Joint Research Project on Priority Areas (YK); the Princess Takamatsunomiya Cancer Research Foundation (YK); Takeda Science Foundation (YK); The Cell Science Research Foundation (YK); Uehara Memorial Foundation (YK and YI); KAKENHI Grant Number 19K09297 (TY), and KAKENHI Grant Number 20K22793 (ZW).

## Author contributions

Y. K. designed the project. Z. W., T. Y., S. I., Y. I., H. K., T. N., H. Z., and Y. K. performed the experiments. S. K., W. H., and S. S. provided critical TIL samples. Y. K. and Z. W. analyzed the data and wrote the manuscript.

## Competing interests

The authors declare the following competing interests: Y. K. received commercial research grants from Takara Bio and Kyowa Kirin. These financial relationships are unrelated to the present study. Y. K. was provided with flow cytometry antibodies by Nippon Becton Dickinson Company as research support through the Outstanding Young Immunology Researcher Award (hosted by Japanese Society for Immunology and Nippon Becton Dickinson Company). All other authors declare no competing interests.
