## [Peer Review File · Communications Biology]

Reviewers' comments:

Reviewer #1 (Remarks to the Author):

The purpose of Wu Z. and colleagues is to identify a novel marker to help distinguish TPEX from other T-cell subsets within the tumor-infiltrating lymphocytes. First the authors show that CAR T cells infused in a cancer mouse model present higher frequency of bona fide TPEX (expressing CCR7 and PD-1) at early timepoints and that these CAR T cells retain higher proliferative capacity. Then they identify CD83 as specific marker for precursor exhausted among the CAR T cells, by employing publicly available gene expression data and flow cytometry. Wu and colleagues then demonstrate that CD83 is induced upon T cell activation with anti-CD3 and anti-CD80. They also nicely investigated the functional significance of CD83 expression and found that it restrains T cell effector functions. Finally, the authors demonstrate that CD83 can identify TPEX even among naturally occurring tumor-infiltrating lymphocytes. The population of progenitor exhausted T cells has been discovered in the past few years. A lot of studies defined this population with very specific markers. The need for additional markers to identify TPEX is not very clear and the author should optimize the focus of their work to increase the relevance of their findings.

The above-described considerations have been reasoned on the following major issues:
Fig. 2c-d: The authors show that Slamf6 is not differentially expressed between terminally exhausted and progenitor exhausted cells. However, as they also state in text, Slamf6 is a marker known to be differentially expressed between the two populations. It would be great if the author could better explain the result.

Fig. 2i: The gating strategy reported here seems a bit confusing. How did the authors set the threshold for CCR7 expression? Showing Ccr7 staining in another sample (i.e., blood, spleen or lymph node) with a clear positive population would help to understand the gate positioning. It seems like this figure is quite fundamental considering that it would show that CD83 is a marker specific for tumor reactive TPEX vs bystander infiltrating T cells.

Fig. 4: The in vitro assay showing that CD83 overexpression restrains production of cytolytic molecules and CD107a is interesting. However, I think that assessing the in vivo functional relevance of this molecule can improve the relevance of these data. Can the authors knock out or overexpress CD83 on T cells in their tumor model?

Fig. 5e-f: Can the authors perform an enrichment analysis with a TPEX signature among the clusters that they identified? This would allow the authors to define the cluster 2 as TPEX in an unbiased manner.

Fig. 6d: Can CD83 be considered a prognostic marker? Is it associated to an improved patient survival?

Reviewer #2 (Remarks to the Author):

In this manuscript, Wu et al. reported CD83 as a novel marker for precursor exhausted T cells (Tpex) using tumor models treated by chimeric antigen receptor (CAR)-engineered T cells, mouse and human tumors.

The major concern is whether CD83 is a really robust marker to define Tpex.

1. Given that the CD83 expression showed the most prominent difference between CCR7-positive and -negative T cell populations (Fig. 2c, d), there should be a very distinct population of CCR7+CD83+PD1+ cells among CD8 T cells. However, as shown in Figure 2i, the CD83 expression is in continuum.

2. The authors should clearly define CD83+ Tpex, as well as Tex and other populations, with strict phenotypic, molecular and functional studies, for instance, the well-defined molecular signatures for Tpex should be presented with own data. However, across the manuscript the authors try to use an indirect way to present the data.

Other suggestions:

- 1) In Figure1C, the result described PD1+CCR7+ CAR-T significantly decreased in later time, but figure Y axis labeling was the frequency of CCR7+ within CAR T+T cells.
- 2) Suggest to add a brief explanation of CD83 in the introduction.
- 3) Figure6C, the number of CD83+CCR7+PD1+ cells was too low, multiple published human scRNAseq datasets should be included to verify the result.

In general, this reviewer feels that this manuscript is not highly convincing, also lacks enough originality.

Reviewer #3 (Remarks to the Author):

The authors identify a surface molecule that characterizes TPEX and demonstrate that CD83, as a potential marker, is predominantly expressed in exhausted T cells using CAR-T cells model. Their work seems interesting, but some issues need to be addressed:

T cell exhaustion exists in very many solid tumors, and the title of this manuscript does not indicate in which tumors are investigated, does this indicate that their results are true in all tumors? If so, why did the authors choose only a limited dataset instead of using more pan-cancer datasets to get more evidence?

Are the surface markers of the exhausted T cells in mice and humans the same? Wherry EJ. Indicates the typical marker for discriminating pre-exhausted T and exhausted T cells is the expression level of TCF (Nat Immunol (2011 Jun; 12(6):492-9. doi: 10.1038/ni.2035.), and in this study, the authors did not detect this marker, please explain it. Or it should be mentioned in the discussion. In addition, the expression differences of LAYN, CXCL13, HAVCR2 and other genes are also markers of exhausted T cells. The author needs to discuss.

Dear Editor:

We respectfully submit our revised manuscript titled, 'CD83 expression characterizes precursor exhausted T cell population.' We thank the Editor and Reviewers for their comments regarding our manuscript. We have revised the manuscript as requested. Please find our point-by-point responses below. The revised sentences of the manuscript are written in red color.

Reviewer #1's comments:

Comment #1.

Fig. 2c-d: The authors show that Slamf6 is not differentially expressed between terminally exhausted and progenitor exhausted cells. However, as they also state in text, Slamf6 is a marker known to be differentially expressed between the two populations. It would be great if the author could better explain the result.

Response to comment #1.

Thank you for the comment. In the revised manuscript, we repeated the screening experiments. For fair comparison among the candidate molecules, we analyzed their expression levels using the same setting: biotin-conjugated mAb followed by streptavidin-PE. As shown in Fig. 2c and 2d, SLAMF6 was positive in almost all the tumor-infiltrating CAR-T cells. There was no significant difference in the frequency of SLAMF6⁺ cells in PD1⁺CCR7⁺, PD1⁺CCR7⁻, and PD1⁻ T cell populations. However, when we compared mean fluorescence intensity of SLAMF6 among the three populations, PD1⁺CCR7⁺ cells showed slightly higher levels of SLAMF6 expression than the other subsets (please see Supplementary Fig. 1 in the revised manuscript).

Preferential expression of Slamf6 in progenitor exhausted T cells was reported in mouse T cells (Utzschneider et al. Immunity 2016; Miller et al. Nat Immunol 2018). One recent study analyzed its expression in human T cells (Hajaj et al. eLife 2020). Although differential expression between progenitor and terminally exhausted T cells was not investigated in that study, SLAMF6 was constitutively expressed in human TIL samples, which is consistent with our data. One possibility is that there is a different transcriptional regulation of SLAMF6 between human and mouse T cells. In addition, our analysis was performed using CAR-T cell treatment models instead of endogenous antitumor T cells. Considering that SLAMF6

expression shows a dynamic change after T cell activation (Hajaj et al. eLife 2020), its expression levels may significantly be affected by target antigen density and avidity of antitumor T cells. We mentioned these possibilities in the Discussion section.

* Page 6, Line 5: Although CD81 and SLAMF6 were mostly expressed in all of the T cell subsets, these molecules also showed increased expression levels in CCR7⁺PD1⁺ T cells compared with the other populations when compared by mean fluorescence intensity (Supplementary Fig. 1).

* Page 14, Line 7: SLAMF6, one of the previously identified T_{PEX} markers in mouse tumor and virus infection models, did not show a marked difference between CCR7⁺ and CCR7⁻ exhausted T cells^{6,8}. One possibility is that there is a different transcriptional regulation between human and mouse T cells. In addition, our analysis was performed using genetically engineered CAR-T cell treatment models, which have substantially different conditions compared with endogenous antitumor T cells within the tumor, including the target antigen density and avidity of the T cells. Considering that SLAMF6 is quickly upregulated after T cell activation⁴⁹, its expression levels may significantly be affected by these parameters.

* Supplementary Fig. 1:

Supplementary Fig. 1. Phenotypic profiles of precursor and terminally exhausted CAR-T-cells. Anti-mesothelin or anti-CD19 CAR-T cells were infused into NSG mice engrafted with A375-mesothelin or A375-CD19, and tumor-infiltrating PD1⁺CCR7^{+/-} or PD1⁻ T cells were analyzed for the indicated surface molecules. The data shown are mean fluorescence intensity of each molecule (n=5 mice, the same samples as shown in Fig. 2d; repeated measures one-way ANOVA with multiple comparison test). * P<0.05, ** P<0.01.

* Fig. 2d:

Fig. 2d. The frequency of cells that express the indicated molecules in the CCR7⁺PD1⁺, CCR7⁺PD1⁻, and PD1⁻ T cell population within the tumor (left panel, n=5 samples, repeated measures one-way ANOVA with multiple comparison test) or in the CCR7^{+/+} peripheral blood T cells (right panel, n=3 samples, two-tailed paired t-test). * P<0.05, ** P<0.01.

Comment #2.

Fig. 2i: The gating strategy reported here seems a bit confusing. How did the authors set the threshold for CCR7 expression? Showing Ccr7 staining in another sample (i.e., blood, spleen or lymph node) with a clear positive population would help to understand the gate positioning. It seems like this figure is quite fundamental considering that it would show that CD83 is a marker specific for tumor reactive TPEX vs bystander infiltrating T cells.

Response to comment #2.

We appreciate this important comment. In the revised manuscript, we always analyzed samples with a clear CCR7⁺ population as comparison. First, we compared the expression of CD83 and other markers among precursor exhausted (CCR7⁺PD1⁺), terminally exhausted (CCR7⁺PD1⁻), and bystander tumor-infiltrating T cells (PD1⁻) in Fig. 2c-d. To determine the threshold for CCR7 expression, we analyzed CD8⁺ T cells of peripheral blood mononuclear cells using the same set of antibodies. As shown in the right panel of Fig. 2c, peripheral blood T cells exhibited a bimodal distribution of CCR7 expression, which enabled us to identify a distinctly CCR7-positive and negative population. Using this gating strategy, we confirmed that CD83 expression was significantly higher in CCR7⁺PD1⁺ CAR-T cells than the other T cell subsets (Fig. 2d).

In the second experiment, we injected CAR-T cells intratumorally and peripheral blood T cells

intravenously into A375-mesothelin-bearing NSG mice and extracted the subcutaneous tumor and spleen. We determined the CCR7[±] threshold according to the plots of T cells in the spleen, which included a population clearly positive for CCR7 (Fig. 3b). We confirmed that CCR7⁺ T cells in the tumor tissue expressed higher levels of CD83 compared with CCR7⁻ TILs and T cells in the spleen.

* Fig. 2c (right panel):

Fig. 2c. Representative flow cytometry plots analyzing the expression of individual molecules in the tumor-infiltrating T cells and peripheral blood T cells. Gating threshold for each molecule was determined based on the plots of the fluorescence minus one (FMO) control samples.

* Fig. 3b:

Fig. 3b. The expression of CD83 was analyzed in CCR7[±] T cells within the tumor and spleen. Representative flow cytometry plots (b) and the frequency of CD83⁺ cells in the CCR7[±]-CD8⁺ T cell populations are shown (c, n=5 mice, repeated measures one-way ANOVA with multiple comparison test).

Comment #3.

Fig. 4: The *in vitro* assay showing that CD83 overexpression restrains production of cytolytic molecules and CD107a is interesting. However, I think that assessing the *in vivo* functional relevance of this molecule can improve the relevance of these data. Can the authors knock out or overexpress CD83 on T cells in their tumor model?

Response to comment #3.

Thank you for the comment. We evaluated the effect of CD83 overexpression on antitumor T cell functions *in vivo*. NSG mice were transplanted with the mesothelin⁺ pancreatic cancer cell line AsPC-1 and treated by mesothelin-targeting CAR-T cells with or without ectopic expression of CD83. We also transduced CAR-T cells with the luciferase gene to longitudinally monitor CAR-T cell persistence.

The mice treated with CD83-overexpressing CAR-T cells initially showed more rapid tumor progression than those with control CAR-T cells (Figure 5k, the data on week 3), which is consistent with the *in vitro* findings showing the attenuated production of cytolytic molecules by CD83-overexpressing T cells. However, CD83-transduced CAR-T cells eventually controlled tumor growth in all the treated mice. When assessed by the luciferase activity, both control and CD83-overexpressing CAR-T cells progressively accumulated in the tumor site at comparable levels. These results suggest that although CD83 overexpression slightly attenuates immediate effector response of CAR-T cells, it does not compromise their long-term antitumor functions. The data were presented in Fig. 5i-l and Supplementary Fig. 4.

* Page 9, Line 8: To investigate the effect of CD83 overexpression on antitumor efficacy of CAR-T cells *in vivo*, we infused mesothelin-targeting CAR-T cells with or without ectopic expression of CD83 into NSG mice that were subcutaneously inoculated with the mesothelin-expressing pancreatic cancer cell line AsPC-1 (Fig. 5i). We also transduced CAR-T cells with the luciferase gene to monitor their persistence. CD83-overexpressing CAR-T cells initially showed less efficient control of tumor growth, which is consistent with the *in vitro* data that ectopic expression of CD83 attenuated the production of cytolytic molecules (Fig. 5j, k). When monitored by the luciferase activity, both control and CD83-overexpressing CAR-T cells progressively accumulated at the tumor site and eventually controlled tumor growth in most of

the treated mice (Fig. 5l; Supplementary Fig. 4).

* Fig 5i-l:

Fig. 5. (i) NSG mice subcutaneously transplanted with the mesothelin⁺ pancreatic cancer cell line AsPC-1 were treated by control or CD83-overexpressing CAR-T cells against mesothelin. CAR-T cells were cotransduced with the luciferase gene to monitor *in vivo* persistence. (j-k) Tumor volume was longitudinally monitored (n=6 mice for each). (l) Total flux of luciferase activity was analyzed by *in vivo* bioluminescent imaging (n=6 mice, repeated measures one-way ANOVA with multiple comparison test for the log-transformed values). The data presented in j-l are a composite of three independent experiments.

* Supplementary Fig. 4:

Supplementary Fig. 4. The effect of CD83 overexpression on *in vivo* antitumor T cell response. NSG mice inoculated with the mesothelin⁺ cell line AsPC-1 were treated by control or CD83-overexpressing CAR-T cells targeting mesothelin. CAR-T cells were transduced with the luciferase gene and longitudinally monitored for *in vivo* persistence and distribution.

Comment #4.

Fig. 5e-f: Can the authors perform an enrichment analysis with a TPEX signature among the clusters that they identified? This would allow the authors to define the cluster 2 as TPEX in an unbiased manner.

Response to comment #4.

Thank you for the important comment. We performed single-cell gene set enrichment analysis among the identified clusters using the singleseqset package (<https://arc85.github.io/singleseqset/index.html>). As shown in Fig. 6g in the revised manuscript, the cluster 2 was significantly enriched with the TPEX-associated gene set that was generated based on the GSE83978 data. These results further validate that the cluster 2 represents a TPEX population.

* Page 11, Line 5: Gene set enrichment analysis among the clusters verified that the TPEX-associated gene set was significantly enriched in the cluster 2 compared with the other clusters (Fig. 6g).

* Page 21, Line 15: Gene set enrichment analysis among the identified clusters was performed using the function “`wmw_gsea`” in the singleseqset R package (version 0.1.2.9000). P-values were adjusted with the false discovery rate. Gene sets associated with memory, precursor exhausted, and terminally exhausted T cells were created by selecting genes with more than 4-fold increased expression than the other populations in the GSE83978 data.

* Fig. 6g:

Fig. 6g. Heatmap of the gene set enrichment analysis for the genes associated with precursor exhausted, terminally exhausted, and memory T cells among the six clusters identified in single-cell RNA-seq analysis. ** P<0.01.

Comment #5.

Fig. 6d: Can CD83 be considered a prognostic marker? Is it associated to an improved patient survival?

Response to comment #5.

Thank you for the comment. In the revised manuscript, we calculated the frequency of CD83⁺CCR7⁺ cells within the CD8⁺ T cell population for individual patients' samples. As shown in Fig. 7e, there was no significant difference in the frequency between responders and non-responders.

We also analyzed several bulk RNA-seq data in which information on clinical response to immune checkpoint inhibitors is available (Supplementary Fig. 7). Although CD83 expression levels were significantly higher in responders than non-responders in some of the datasets, there was no significant difference in the survival of patients between the CD83^{high} and CD83^{low} groups. These results suggest that the analysis of CD83 expression alone does not help to predict survival of the patients.

* Page 12, Line 7: The frequency of CD83⁺CCR7⁺ cells analyzed for individual patients' samples was not significantly different between responders and non-responders to ICI (Fig. 7e). We further analyzed multiple bulk RNA-seq data with prognostic information³³⁻³⁶. Although CD83 expression was related to therapeutic response to ICI in some of the data, its high expression was not significantly associated with survival of the patients, suggesting that CD83 expression alone cannot predict overall prognosis of the patients treated with ICI (Supplementary Fig. 7).

* Fig. 7e:

Fig. 7e. The frequency of CD83⁺CCR7⁺ cells was individually calculated in the samples from patients with or without response to immune checkpoint inhibitors using the GSE120575 dataset (n=9 or 10, unpaired two-tailed *t*-test; NS, not significant).

* Supplementary Fig. 7:

Supplementary Fig. 7. The prognostic impact of CD83 expression levels in patients treated with immune checkpoint inhibitors. (a-e) Patients with melanoma (a-c) or lung cancer (d, e) treated with immune checkpoint inhibitors (ICI) were analyzed for the association of CD83 expression levels with prognosis using the indicated datasets. In **a** and **d**, CD83 expression levels were compared between responders and non-responders to ICI (two-tailed unpaired t-test). In **b**, **c**, and **e**, overall survival or progression-free survival was compared between patients grouped according to the expression levels of CD83 (log-rank test). The median expression value of *CD83* in each cohort was used as a cut-off value. NS, not significant.

Reviewer #2's comments:

Comment #1.

Given that the CD83 expression showed the most prominent difference between CCR7-positive and -negative T cell populations (Fig. 2c, d), there should be a very distinct population of CCR7+CD83+PD1+ cells among CD8 T cells. However, as shown in Figure 2i, the CD83 expression is in continuum.

Response to comment #1.

Thank you for the important comment. To improve the detection resolution of CD83 expression in flow cytometry analysis, we compared anti-CD83 antibodies with different conjugates. As shown in the Figure below, staining with biotin-conjugated anti-CD83 mAb followed by streptavidin-PE most clearly discriminated between stimulated and unstimulated CAR-T cells.

Figure for review purpose only. CD19-targeting CAR-T cells were rested or stimulated with NALM-6 (CD19⁺). The expression of CD83 was analyzed with flow cytometry using anti-CD83 antibodies with the indicated fluorochromes.

We thus evaluated CD83 expression in intratumoral CAR-T cells using the biotin-conjugated antibody in the revised manuscript (Figure 2c-d and Figure 3). We presented FACS plots showing the distribution of CD83 and CCR7 expression in Fig. 3b. The CCR7⁺ intratumoral CAR-T cells were clearly positive for CD83 compared with CCR7⁺ T cells in the spleen. The CCR7⁻ population within the tumor contained both CD83⁺ and CD83⁻ cells.

As the review pointed out, the expression of CD83 still showed continuous, rather than bimodal, distribution patterns, and the CD83 expression analysis alone would not be sufficient to identify a distinct T_{PEX} population. However, the CD83/CCR7 double positive population can be clearly identified and possess functional properties consistent with T_{PEX} (please also see

our response to the comment #2 for functional assays).

* Fig. 3b:

Fig. 3b, c. The expression of CD83 was analyzed in CCR7⁺ T cells within the tumor and spleen. Representative flow cytometry plots (b) and the frequency of CD83⁺ cells in the CCR7⁺CD8⁺ T cell populations are shown (c, n=5 mice, repeated measures one-way ANOVA with multiple comparison test).

* Page 6, Line 13: To examine the expression of CD83 in tumor-infiltrating CAR-T cells compared with T cells outside the tumor, we administered anti-mesothelin CAR-T cells intratumorally into subcutaneous A375-mesothelin tumors and uncultured T cells from the same donor intravenously (Fig. 3a). When analyzed on day 5 after T-cell injection, CD83 was more upregulated in the intratumoral CCR7⁺ CAR-T cell population than in CCR7⁻ CAR-T cells in the tumor and T cells in the spleen (Fig. 3b, c).

* Page 7, Line 15: These results collectively suggest that T cells double positive for CD83 and CCR7 possess functional properties of precursor exhausted T cells .

Comment #2.

The authors should clearly define CD83⁺ T_{pex}, as well as T_{ex} and other populations, with strict phenotypic, molecular, and functional studies, for instance, the well-defined molecular signatures for T_{pex} should be presented with own data. However, across the manuscript the

authors try to use an indirect way to present the data.

Response to comment #2.

Thank you for the essential comment. In the revised manuscript, we analyzed multiple phenotypic and functional properties of CD83⁺ T cells compared with CD83⁻ T cells within the tumor. First, we analyzed TCF7 and granzyme B expression in the intratumoral CAR-T cells. As presented in Fig. 3f-i, CD83⁺ T cells expressed higher levels of TCF7 and, conversely, lower levels of granzyme B than CD83⁻ T cells. These results are consistent with previously described phenotypes of T_{PEX}. Although we also examined the expression of TIM3, which is reported to be predominantly expressed in terminally exhausted T cells compared with T_{PEX}, we did not observe a substantial difference among the three subsets within the PD1⁺ CAR-T cells (CCR7⁺CD83⁺, CCR7⁻CD83⁺, and CD83⁻).

Next, we investigated functional properties of CD83⁺ CAR-T cells. Since we could not obtain enough number of T cells from the tumor for functional assays, we used CAR-T cells repeatedly stimulated *in vitro*, which mimics T cell exhaustion (Belk et al. Cancer Cell 2022; Good et al. Cell 2021). We isolated three populations (CD83⁺CCR7⁺, CD83⁻CCR7⁺, and CD83⁻ cells) and restimulated them *in vitro* (Fig. 3k). As shown in Fig. 3l-m, CD83⁺CCR7⁺ CAR-T cells displayed superior proliferation and IL-2 production compared with the other subsets, which are previously described attributes of precursor exhausted T cells. The secretion of IFN- γ was not significantly different among the three subsets.

These results collectively suggest that precursor exhausted T cells can be detected in CD83⁺ T cells and further enriched in the CD83⁺CCR7⁺ T cell population.

* Fig. 3d-i:

Fig. 3d-i. NSG mice subcutaneously inoculated with the A375-mesothelin were infused with mesothelin-targeting CAR-T cells intratumorally and peripheral blood T cells from the same donor intravenously. T cells within the tumor and spleen were analyzed 5 days after infusion.

(d, e) The expression of CCR7 and TIM3 was compared between CD83^{+/−}PD1⁺CD8⁺ CAR-T cells within the tumor. Representative flow cytometry plots (d) and the frequency of CCR7⁺ cells in each T cell subset are shown (e, n=5 mice, paired two-tailed t-test). The data in c and e are derived from the same samples. (f-i) Expression levels of TCF7 and granzyme B of intratumoral CAR-T cells were analyzed by intracellular flow cytometry. The data shown are representative flow cytometry plots (f, h) and the mean fluorescence intensity of TCF7 (g) and granzyme B (i) in the CD83⁺ and CD83[−] CAR-T cell populations. (n=6 or 5 mice, paired two-tailed t-test).

Fig. 3j-n. Mesothelin-targeting CAR-T cells were daily stimulated with K562-mesothelin for 7 days, and CCR7⁺CD83⁺, CCR7[−]CD83⁺, and CD83[−]CD8⁺ CAR-T cells were purified by flow cytometry (k, CD83-transduced cultured T cells were analyzed using the same panel to determine gating threshold). The isolated T cells were then restimulated by K562-mesothelin to analyze fold expansion (l, n=6) and the secretion of IL-2 (m, n=5; two samples from CD83⁺CCR7[−] and four samples from CD83[−] cells were under detection limit) and IFN-γ (n, n=5; one sample from CD83[−] cells was under detection limit). In l-n, statistical significance was tested by repeated measures one-way ANOVA with multiple comparison test. For m and n, log-transformed values were used for calculation. NS, not significant.

Page 6, Line 18: The expression of CCR7 was selectively observed in the CD83⁺PD1⁺ cells but not in the CD83⁻PD1⁺ T cell population (Fig. 3d, e). Both CCR7⁺ and CCR7⁻ CAR-T cells similarly expressed TIM3 at high levels compared to T cells in the spleen. Further phenotypic analysis showed that CD83⁺ CAR-T cells expressed increased levels of TCF7 and, conversely, decreased levels of granzyme B compared to CD83⁻ CAR-T cells (Fig. 3f-i). These phenotypic features are consistent with those of previously described precursor exhausted T cells^{6,22,23}. We further investigated functional properties of CD83⁺ CAR-T cells. To obtain enough cell numbers, we exploited an *in vitro* chronic stimulation protocol that was reported to induce dysfunctional T cells mimicking exhausted T cells (Fig. 3j)^{24,25}. Since CD83⁺ T cells included both CCR7⁺ and CCR7⁻ cells, and CD83⁻ cells were mostly negative for CCR7 as was seen in the tumor model, we isolated three populations (CD83⁺CCR7⁺, CD83⁻CCR7⁺, and CD83⁻ cells) and restimulated them *in vitro* (Fig. 3k). As shown in Fig. 3l and m, CD83⁺CCR7⁺ CAR-T cells displayed superior proliferation and IL-2 production compared with the other subsets, which are previously reported features of precursor exhausted T cells^{6,8,10}. These results collectively suggest that T cells double positive for CD83 and CCR7 possess functional properties of precursor exhausted T cells.

Comment #3.

In Figure 1C, the result described PD1⁺CCR7⁺ CAR-T significantly decreased in later time, but figure Y axis labeling was the frequency of CCR7⁺ within CAR T⁺T cells.

Response to comment #3.

We apologize for the discrepancy. In the revised manuscript, we presented the frequency of PD1⁺CCR7⁺ cells within CAR-T cell population, which was significantly lower in the samples collected at later time points.

* Fig. 1c:

Fig. 1c. The frequency of **PD1⁺CCR7⁺** cells in the intratumoral CD8⁺ CAR-T cells at early (day 14 or 21) or late (day 26 or 34) time points (n=6, unpaired two-tailed *t*-test).

Comment #4.

Suggest to add a brief explanation of CD83 in the introduction.

Response to comment #4.

Thank you for the comment. We explained about CD83 in the Introduction section.

* Page 4, Line 8: **CD83 is a member of the immunoglobulin superfamily and highly expressed in mature dendritic cells to function as one of the costimulation and adhesion molecules²⁰. Although activated T cells also upregulate CD83 expression²¹, its expression dynamics has not been elucidated in detail. We demonstrate that CD83 is preferentially upregulated in less-differentiated T cells upon antigen stimulation.**

Comment #5.

Figure 6C, the number of CD83+CCR7+PD1+ cells was too low, multiple published human scRNAseq datasets should be included to verify the result.

Response to comment #5.

Thank you for the important suggestion. In the revised manuscript, we analyzed scRNA-seq data derived from GSE156728 (pan-cancer analysis), GSE98638 (hepatocellular carcinoma), GSE115978 (melanoma), and GSE190202 (breast cancer) in addition to GSE120575 (melanoma). As shown in Figure 7c, CD83-expressing cells were significantly enriched in the

CCR7⁺PD1⁺ cell population compared with the CCR7⁻PD1⁺ or PD1⁻ T cell subsets throughout different datasets.

To further validate this classification approach, we compared the frequency of HAVCR2 (TIM3)⁺ T cells within these three populations. In contrast to CD83, the CCR7⁻PD1⁺ T cell population included TIM3⁺ cells more frequently than the other populations, which is consistent with the previous findings that TIM3 is highly expressed in the terminally exhausted T cell population.

* Page 11, Line 13: We also analyzed the publicly available single-cell RNA-seq dataset of TIL samples derived from various cancer types: GSE156728 (pan-cancer)²⁸, GSE98638 (hepatocellular carcinoma)²⁹, GSE115978 (melanoma)³⁰, GSE120575 (melanoma)³¹, and GSE190202 (breast cancer)³². Overall, the frequency of CD83-expressing cells was significantly higher in the CD8⁺ T cells that expressed both PD1 and CCR7 than in the CCR7⁻PD1⁺ or PD1⁻ T cells, which is consistent with the above results (Fig. 7c). In contrast, *HAVCR2* (TIM3)-expressing T cells were significantly more abundant in the CCR7⁻PD1⁺ population than the other subsets.

* Page 22, Line 2: Single-cell RNA-seq data of TIL (GSE156728²⁸, GSE98638²⁹, GSE115978³⁰, GSE120575³¹, and GSE190202³²) were analyzed using the raw or normalized count files retrieved from the gene expression omnibus (GEO) database. Non-zero read counts were defined as positive expression. For the data from GSE98638, GSE115978, and GSE120575, CD8⁺ T cells were selected based on the expression of *CD8A* (transcripts per million [TPM] values >3).

* Fig. 7c:

Fig. 7c. Single-cell RNA-sequencing data of TIL from various cancer types were analyzed for the proportion of CD83⁺ cells within the CCR7⁺PD1⁺, CCR7⁻PD1⁺, and PD1⁻ CD8⁺ T cell populations (n=12 datasets, repeated measures one-way ANOVA with multiple comparison test).

Reviewer #3's comments:

Comment #1.

T cell exhaustion exists in very many solid tumors, and the title of this manuscript does not indicate in which tumors are investigated, does this indicate that their results are true in all tumors? If so, why did the authors choose only a limited dataset instead of using more pan-cancer datasets to get more evidence?

Response to comment #1.

Thank you for the important comment. In the revised manuscript, we retrieved pan-cancer single-cell RNA-seq data from GSE156728 (Zheng et al. Science 2021). We calculated the frequency of CD83⁺ cells in the CCR7⁺PD1⁺ (precursor exhausted), CCR7⁻PD1⁺ (terminally exhausted), and PD1⁻ (bystander) T cells. In addition, we also obtained data from GSE98638 (hepatocellular carcinoma), GSE115978 (melanoma), and GSE190202 (breast cancer). Similar to the results shown in the original manuscript using the melanoma dataset (GSE120575), CD83-positive cells were enriched in the CCR7⁺PD1⁺ cell population in most of the cancer types (Fig. 7c). As comparison, we also calculated the frequency of *HAVCR2* (TIM3)⁺ T cells within these three populations. Consistent with the previous findings that TIM3 is predominantly upregulated upon terminal exhaustion, the CCR7⁻PD1⁺ T cells included TIM3⁺ cells more frequently than the other populations. These results suggest that the preferential expression of CD83 in precursor exhausted T cells is not limited to a specific cancer type.

* Page 11, Line 13: We also analyzed the publicly available single-cell RNA-seq dataset of TIL samples derived from various cancer types: GSE156728 (pan-cancer)²⁸, GSE98638 (hepatocellular carcinoma)²⁹, GSE115978 (melanoma)³⁰, GSE120575 (melanoma)³¹, and GSE190202 (breast cancer)³². Overall, the frequency of CD83-expressing cells was significantly higher in the CD8⁺ T cells that expressed both PD1 and CCR7 than in the CCR7⁻PD1⁺ or PD1⁻ T cells, which is consistent with the above results (Fig. 7c). In contrast, *HAVCR2* (TIM3)-expressing T cells were significantly more abundant in the CCR7⁻PD1⁺ population than the other subsets.

* Page 22, Line 2: Single-cell RNA-seq data of TIL (GSE156728²⁸, GSE98638²⁹, GSE115978³⁰,

GSE120575³¹, and GSE190202³²) were analyzed using the raw or normalized count files retrieved from the gene expression omnibus (GEO) database. Non-zero read counts were defined as positive expression. For the data from GSE98638, GSE115978, and GSE120575, CD8⁺ T cells were selected based on the expression of *CD8A* (transcripts per million [TPM] values >3).

* Fig. 7c:

Fig. 7c. Single-cell RNA-sequencing data of TIL from various cancer types were analyzed for the proportion of CD83⁺ cells within the CCR7⁺PD1⁺, CCR7⁻PD1⁺, and PD1⁻ CD8⁺ T cell populations (n=12 datasets, repeated measures one-way ANOVA with multiple comparison test).

Comment #2.

Are the surface markers of the exhausted T cells in mice and humans the same? Wherry EJ. Indicates the typical marker for discriminating pre- exhausted T and exhausted T cells is the expression level of TCF (Nat Immunol. 2011 Jun; 12(6):492-9. doi: 10.1038/ni.2035.), and in this study, the authors did not detect this marker, please explain it. Or it should be mentioned in the discussion.

Response to comment #2.

Thank you for the comment. In the revised manuscript, we analyzed the expression of TCF7 and CD83 in tumor-infiltrating CAR-T cells by intracellular flow cytometry. As shown in Fig. 3f and g, CD83⁺ CAR-T cells showed significantly higher levels of TCF7 expression compared with the CD83⁻ T cell population. These results further support that CD83 marks precursor exhausted T cells. We also added the above review paper to the references.

* Page 7, Line 2: Further phenotypic analysis showed that CD83⁺ CAR-T cells possessed increased expression of TCF7 and, conversely, decreased expression of granzyme B compared to CD83⁻ CAR-T cells (Fig. 3f-i). These phenotypic features are consistent with those of previously described precursor exhausted T cells^{6,22,23}.

* Fig. 3f, g:

Fig. 3f-i. Expression levels of TCF7 and granzyme B of intratumoral CAR-T cells were analyzed by intracellular flow cytometry. The data shown are representative flow cytometry plots (f, h) and the mean fluorescence intensity of TCF7 (g) and granzyme B (i) in the CD83⁺ and CD83⁻ CAR-T cell populations. (n=6 or 5 mice, paired two-tailed t-test).

Comment #3.

In addition, the expression differences of LAYN, CXCL13, HAVCR2 and other genes are also markers of exhausted T cells. The author needs to discuss.

Response to comment #3.

Thank you for the comment. As the reviewer pointed out, multiple genes have been reported to be differentially expressed among exhausted T cell population. For HAVCR2 (TIM3), we analyzed its expression in our tumor model. As shown in Fig. 3d, its expression levels were comparable between PD1⁺CCR7⁺ and PD1⁺CCR7⁻ T cells. Both populations highly expressed TIM3 compared with T cells in the spleen. These data are not consistent with the previous studies showing that TIM3 expression marks terminally exhausted T cells. We mentioned these results in the Discussion section. We also discussed other exhaustion markers in the Discussion section.

* Fig. 3d:

Fig. 3d, e. The expression of CCR7 and TIM3 was compared between CD83⁺PD1⁺CD8⁺ CAR-T cells within the tumor. Representative flow cytometry plots (d) and the frequency of CCR7⁺ cells in each T cell subset are shown (e, n=5 mice, paired two-tailed t-test).

* Page 14, Line 3: Several other molecules have recently been reported to have predominant expression in a subset of exhausted T cells, which include CXCL13^{28,47}, LAYN⁴⁸, and CD69³⁸. Combinatorial use of these markers may further enhance detection of tumor-reactive T cells with superior survival potential.

* Page 14, Line 7: SLAMF6, one of the previously identified T_{PEX} markers in mouse tumor and virus infection models, did not show a marked difference between CCR7⁺ and CCR7⁻ exhausted T cells^{6,8}. One possibility is that there is a different transcriptional regulation between human and mouse T cells. In addition, our analysis was performed using genetically engineered CAR-T cell treatment models, which have substantially different conditions compared with endogenous antitumor T cells within the tumor, including the target antigen density and avidity of the T cells. Considering that SLAMF6 is quickly upregulated after T cell activation⁴⁹, its expression levels may significantly be affected by these parameters. Similarly, although previous studies demonstrated a predominant expression of TIM3 in terminally exhausted T cells⁶⁻¹², we did not observe a prominent difference in the expression levels of TIM3 between CD83⁺ CAR-T cells in our models.

REVIEWERS' COMMENTS:

Reviewer #1 (Remarks to the Author):

We appreciate the effort the authors undertook to address the critique and suggestions we posed for "CD83 expression characterizes precursor exhausted T cell population" by Wu et al. and the newly added data answered our major concerns. After reviewing again their manuscript, the authors may now want to answer some remaining minor suggestions. We hope these can help them to make the overall message even clearer.

Page 4, lines 7-8 and 11-12: it would probably help the readers if the authors could improve the consistency of these two sentences.

Page 7, line 2: "Both CCR7+ and CCR7- CAR-T cells similarly expressed TIM3 at high levels compared to T cells in the spleen." It would be great to have a graph summarizing the authors' observation alongside the representative flow cytometry plots.

Page 12, line 5-8: The results presented in Fig. 7d and in Fig. 7e seem to be in contrast. The authors should state clearer what are differences between the two data and what is the result.

Reviewer #2 (Remarks to the Author):

The authors have nicely addressed all my concerns.

Reviewer #3 (Remarks to the Author):

This revised version seems better than before and answer my concerns.

Response Letter

Reviewer #1's comments:

Comment #1.

Page 4, lines 7-8 and 11-12: it would probably help the readers if the authors could improve the consistency of these two sentences.

Response to comment #1.

Thank you for the comment. We have revised this paragraph to explain about the aim and results of the present study concisely.

* Page 4, Line 7-12: In this study, we aimed to identify a surface molecule that characterizes TPEX and demonstrate that CD83 is predominantly expressed in exhausted T cells with an early memory phenotype. CD83 is a member of the immunoglobulin superfamily and highly expressed in mature dendritic cells to function as one of the costimulation and adhesion molecules²⁰. Although activated T cells also upregulate CD83²¹, its expression dynamics has not been elucidated in detail. **We characterize the expression kinetics and functional roles of CD83 in antitumor T cells.**

Comment #2.

Page 7, line 2: "Both CCR7⁺ and CCR7⁻ CAR-T cells similarly expressed TIM3 at high levels compared to T cells in the spleen." It would be great to have a graph summarizing the authors' observation alongside the representative flow cytometry plots.

Response to comment #2.

We appreciate this important comment. In the revised manuscript, we have shown mean fluorescence intensity of TIM3 in the four populations presented in Figure 3d: CD83⁺PD1⁺CCR7⁺, CD83⁺PD1⁺CCR7⁻, and CD83⁻PD1⁺CCR7⁻ T cells in the tumor and PD1⁻ T cells in the spleen. As stated in the main text, both CCR7⁺ and CCR7⁻ CAR-T cells in the tumor showed higher expression levels of TIM3 than those in the spleen. These data were shown in Figure 3f of the revised manuscript.

* Fig. 3f:

Fig. 3. (d–f) The expression of CCR7 and TIM3 was compared between CD83^{+/−}PD1⁺CD8⁺ CAR-T cells within the tumor. The data shown are representative flow cytometry plots (d), the frequency of CCR7⁺ cells in CD83^{+/−}PD1⁺ CAR-T cells in the tumor (e, n=5 mice, paired two-tailed t-test), and **mean fluorescence intensity of TIM3 in the indicated T cell populations (f, n=5 mice, repeated measures one-way ANOVA with multiple comparison test).**

Comment #3.

Page 12, line 5-8: The results presented in Fig. 7d and in Fig. 7e seem to be in contrast. The authors should state clearer what are differences between the two data and what is the result.

Response to comment #3.

We apologize for the confusing description. In the revised manuscript, we explained about how these data were generated in more detail. In Figure 7d, the frequency of CD83⁺CCR7⁺ cells was calculated by summing the total number of cells included in the group (responder or non-responder at baseline or after ICI treatment). In Figure 7e, we calculated the frequency for individual patients' samples.

* Page 12, line 4: When we counted the total number of CD83⁺CCR7⁺ cells in each of the four groups (pre- or posttreatment melanoma samples from responders or non-responders to ICI), the proportion of CD83⁺CCR7⁺ cells within the CD8⁺ T cell population was higher in the responder group (43 of 1,005 cells, 4.3%) than in the non-responder group (26 of 1,587 cells, 1.6%) at baseline, which significantly decreased after ICI treatment (25 of 1082 cells, 2.3%) (Fig. 7d). However, the frequency of CD83⁺CCR7⁺ cells calculated for individual patients' samples was not significantly different between responders and non-responders to ICI (Fig. 7e). These results might be due to the insufficient number of patients or because CD83 has poor predictive power when used as a single marker.

Reviewer #2's comments:

The authors have nicely addressed all my concerns.

Reviewer #3's comments:

This revised version seems better than before and answer my concerns.

Response to the comments.

Thank you for the comment. The reviewers' suggestions greatly helped to improve our manuscript.